# Identifying Outcome-oriented Root Causes via Cross Regression

## Abstract

Root Cause Analysis (RCA) in complex and interconnected systems exhibits significant importance in fields such as microservice maintenance, and supply-chain management. By identifying every intervened variable, existing RCA methods have achieved remarkable progress in localizing and fixing anomalies. However, people may be more interested and focused on those intervened variables that produced effects on a specific outcome, rather than the intervened variables that do not necessarily affect that outcome. This raises concerns on redundant localizing and extra efforts in fixing the anomalies. To fill this gap, we study a novel and challenging problem, termed as **O**utcome-**O**riented **R**oot-**C**ause **A**nalysis (OORCA), aiming to identify all **intervened ancestor variables** of the outcome variable. To handle the proposed OORCA problem, we then propose the **C**ross-**R**egressing-based **R**oot **C**ause (CRRC) framework by cross-regressing observational (normal) and interventional (abnormal) data on the outcome variable. Theoretically, our identifiability analysis proves that the proposed CRRC can capture all outcome-oriented root causes, and our asymptotic analysis offers tractable and informative criteria in the finite-sample regime. Extensive experiments on four benchmarks with 13 competitive baselines highlight the superiority of CRRC in both accuracy and running efficiency.

## 1 Introduction

Detecting anomalies in operational systems plays a crucial role in various domains, such as cloud services (Li et al., 2022; Shan et al., 2019), medical monitoring (Strobl, 2024), industrial manufacturing (Susto et al., 2017), and smart city governance (Budhathoki et al., 2022). **Root-Cause Analysis (RCA)** (Li et al., 2022; Shan et al., 2019; Budhathoki et al., 2022; Ikram et al., 2022), which identifies the underlying causes of anomalies, is particularly important for effective localization and resolution, and has become increasingly needed as the system complexity grows. For example, considering the example graph in Fig.1 (a), with $X$ served as variables in a running system (e.g., processing nodes in the supply-chain), $Y$ served as a outcome variable of interest (e.g., final received orders), and edges as causal dependencies across variables.

Unfortunately, the causal graph characterizing the dependencies among variables is *unknown in prior* (Pearl, 2009), which poses significant challenges. Thereby, it is imperative to localize the root causes *solely from the observational data*, which encompasses both normal and abnormal observations. Benefiting from the concept of causal intervention, **existing RCA methods consider that the outcome anomaly** is caused by some interventions on other variables, so that normal and abnormal observations are obtained from the natural distribution and the intervened distribution, respectively (Ikram et al., 2022; Li et al., 2022; Pearl, 2009; Varici et al., 2021). Based on both types of data, these RCA methods aim to identify the set of *all intervened variables* (Li et al., 2022; Ikram et al., 2022; Pham et al., 2024) *without acquiring the prior causal structures*, which includes three categories: (i) discovery-based methods, which augments and identify the (partially) underlying causal graph with an additional "intervention" node (Ikram et al., 2022; Zheng et al., 2024; Yang et al., 2024); (ii) counterfactual reasoning methods, which models the happens of anomaly as interventions only on noise terms (Budhathoki et al., 2022; Okati et al., 2024; anonymous, 2025); and (iii) statistical methods (Varici et al., 2021; Chen et al., 2024), which checks whether the pre-constructed statistics, e.g., entries of the precision matrix (Varici et al., 2021), exhibits shifts.

However, instead of identifying all intervened variables, people may be more interested and focused on intervened variables with causal effects on the outcome variable of interest. As shown in Fig. 1 (b), when anomalies of "$Y$" happens, a bunch of existing RCA methods identify both "$X_1$" and "$X_2$" as root causes, while "$X_2$" contributes zero to anomalies happened on the outcome. Consequently, with a large amount of variables (e.g., a bunch of "$X_2$") exhibited in realistic running systems (Budhathoki et al., 2022; Ikram et al., 2022), localizing and fixing thousands of such unneces-

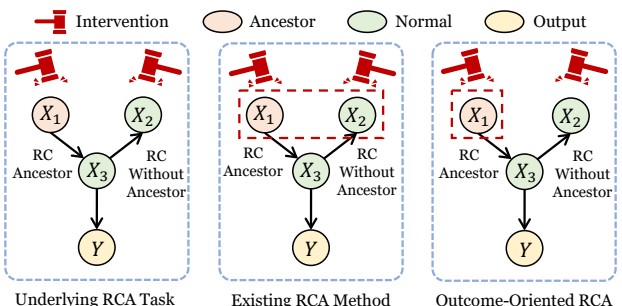

Figure 1: Supply-chain example of: (a) **Left.** Data of the RCA Task; (b) **Center.** Traditional RCA methods; (c) **Right.** Our OORCA Problem.

sary variables raises significantly redundant fixing efforts. Although some RCA have attempted to identify intervened ancestors as root causes (e.g., $X_1$) (Budhathoki et al., 2022; Okati et al., 2024; anonymous, 2025), their prior knowledge requirements such as a known causal graph or accessible Structural Causal Model (SCM) become restrictive in realistic applications.

To address this gap, we propose identifying all **intervened ancestors** of the outcome variable among upstream variables (Fig. 1 (c)), termed **O**utcome-**O**riented **R**oot-**C**ause **A**nalysis (OORCA), *without prior knowledge of the causal graph*. Concretely, we introduce **C**ross **R**egressing-based **R**oot **C**ause Identification (CRRC), a two-stage procedure: (1) identify ancestors of $Y$ by regressing $X$ on non-linear transforms of $Y$ (e.g., $X_1$ or $X_3$); (2) for each ancestor $X_k$ (e.g., $X_1, X_3$), test its necessity by comparing two regression coefficients, i.e., one from observational data, the other replacing $X_k$ with its interventional counterpart, thus the outcome-oriented root causes (e.g., $X_1$) are separately identified. Theoretically, CRRC captures all outcome oriented root causes and ensures asymptotic normality of coefficient differences, enabling hypothesis testing for root-cause identification.

We summarize our contributions as follows:

- We study a novel and challenging problem named **Outcome-Oriented Root-Cause Analysis (OORCA)**, by identifying all intervened ancestors to the outcome variable, *without prior knowledge of causal graph*.

- To solve this problem, we then contribute a novel regression-driven method termed CRRC by identifying ancestors and comparing regression coefficient.

- Theoretically, we prove that our CRRC is capable to capture all necessary root-causes affecting the outcome variable, together with the asymptotic normality analysis, which supports the hypothesis testing on coefficient differences.

- Extensive experiments show that our CRRC surpasses 13 baseline methods by a large margin on both synthetic, semi-synthetic and real-world benchmarks.

## 2 PROBLEM DEFINITION

**Notations.** Throughout this paper, we use the upper-case letter, e.g., $X$, to denote the random variables, with the lower-case letter, e.g., $x$, to represent the corresponding realizations. We then associate a Directed Acyclic Graph (DAG) $\mathcal{G} = ([p+1], E)$ with the node set $[p+1] \triangleq \{1, 2, \ldots, p+1\}$ (we use the index $p+1$ to denote the outcome variable variable $Y$) and the edge set $E \subseteq [p+1] \times [p+1]$, where the directed edge from $i \in [p+1]$ to $j \in [p+1]$ is denoted as $i \to j$. We use the notations $Pa(X^*)$, $Ch(X^*)$, $An(X^*)$ and $De(X^*)$ to denote the set of parents, children, ancestors and descendant of variable $X^*$, respectively[1]. Furthermore, we term any directed path from $X^*$ to variables in $De(X^*)$, e.g., $X^* \to X_1 \to \cdots \to X_{De}^*$, $X_{De}^* \in De(X^*)$, as the *causal path*. Besides, we denote the $k$-th row and column of a matrix $x$ as $x_{k.}$ and $x_{.k}$, respectively. Throughout

---

[1]We allow $X \in Pa(X)$ and $X \in An(X)$ in this paper.

this paper, we use $\tilde{x}^k$ and $\tilde{X}^k$ to denote the empirical data and the corresponding random vector after substituting the $k$-th feature $X_{.k}$ by its interventional column, that is, $\tilde{X}_{.k}$.

**Data Generation Process.** Following (Varici et al., 2021; Li et al., 2024; Shimizu, 2014), we describe the joint generation of observed samples $X = \{X_1, X_2, \ldots, X_p\}$ and the outcome variable $Y$ with a linear structural causal model (SCM):

$$\begin{pmatrix} X \\ Y \end{pmatrix} = G \begin{pmatrix} X \\ Y \end{pmatrix} + Q \odot \Phi, \tag{1}$$

where the vector of exogenous noise term $\Phi = \{\Phi_1, \Phi_2, \ldots, \Phi_p, \Phi_Y\}$, $G \in \mathcal{R}^{(p+1) \times (p+1)}$ is an autoregressive matrix, i.e., $G_{ij} \neq 0$ if and only if $j \rightarrow i$ in $\mathcal{G}$, and the vector $Q \in \mathcal{R}^{p \times 1}$ characterizes the effects of $\Phi_i$. We assume that the SCM is Markovian such that $\{\Phi_1, \Phi_2, \ldots, \Phi_p, \Phi_Y\}$ are independent variables. Besides, we assume that each $\Phi$ is centered with finite variance, i.e., $\mathbb{E}[\Phi_i] = 0$ and $\epsilon_i = \text{var}(\Phi_i) < \infty$.

**Anomalies Modeling.** Following previous observations and foundations Ikram et al. (2022); Chen et al. (2024), we model anomalies observed on $Y$ as the results of soft interventions Kocaoglu et al. (2019); Chen et al. (2024); Ikram et al. (2022); Varici et al. (2021) on the system comprised of $X$:

**Definition 1** (Soft Interventions). *$X_i$ is soft-intervened, if the mechanism (conditional) of each $X_i$ is replaced with another mechanism, i.e., $P(X_i \mid Pa(X_i)) \rightarrow \tilde{P}(X_i \mid Pa(X_i))$.*

With the introduced notations and characterizations of anomalies, we now present our formal definitions on outcome-oriented root-cause analysis.

**Definition 2** (Outcome-Oriented Root-cause Analysis (OORCA)). *Without acquiring the SCM in Eq.(1) or the causal DAG $\mathcal{G}$, given mixed empirical data, including the observational (normal) data as $\mathcal{D} = (x \in \mathcal{R}^{n \times p}, y \in \mathcal{R}^{n \times 1})$ and the interventional (abnormal) data $\tilde{\mathcal{D}} = (\tilde{x} \in \mathcal{R}^{n \times p}, \tilde{y} \in \mathcal{R}^{n \times 1})$, our OORCA aims to identify the necessary root-cause set $X^{RC}$ satisfying the following conditions:*

*(1) $\forall X \in X^{RC}$, $X \in An(Y)$;*

*(2) $\forall X \in X^{RC}$, $X$ is soft-intervened as in Def. 1.*

**Distinction from Previous RCA Setup.** Previous RCA methods, i.e., which target at identifying all intervened variables, satisfy only condition (2) in our Def. 2 (Li et al., 2022; Ikram et al., 2022; Chen et al., 2024; Varici et al., 2021). By contrast, our OORCA problem targets at a more finer-grained but challenging identification result. Besides, we also note that some concurrent researches (Nagalapatti et al., 2025; Budhathoki et al., 2022) have also exhibited aligned objectives of conditions (1) and (2) in our Def.2. *However, they concentrate on counterfactual-based RCA, requiring tractable SCMs or prior causal graphs, which differs from the fundamental setup of our OORCA problem.*

**Remark 1** (Difference from Anomaly Detection). *Anomaly detection refers to the detection of drastic shifts on the outcome variable, i.e., $Y$; while RCA aims to localize a subset of upstream variables in $X$ that causing the anomaly of $Y$. In summary, as RCA and anomaly detection target at different stages with divergent areas, our approach is only triggered when an anomaly of $Y$ is detected;*

## 3 OUR METHOD: CROSS-REGRESSION FOR THE OORCA PROBLEM

In this section, we first adopt an ancestor mining strategy (Schultheiss & Bühlmann, 2023) that offers efficient support for identifying ancestors relevant to the outcome variable, and then detail the core component of our method, i.e., the proposed cross-regression algorithm.

### 3.1 PRE-STEP: IDENTIFYING ANCESTORS WITH REGRESSION

We implement the pre-step of our CRRC by the ancestor regression method based on the following established lemma:

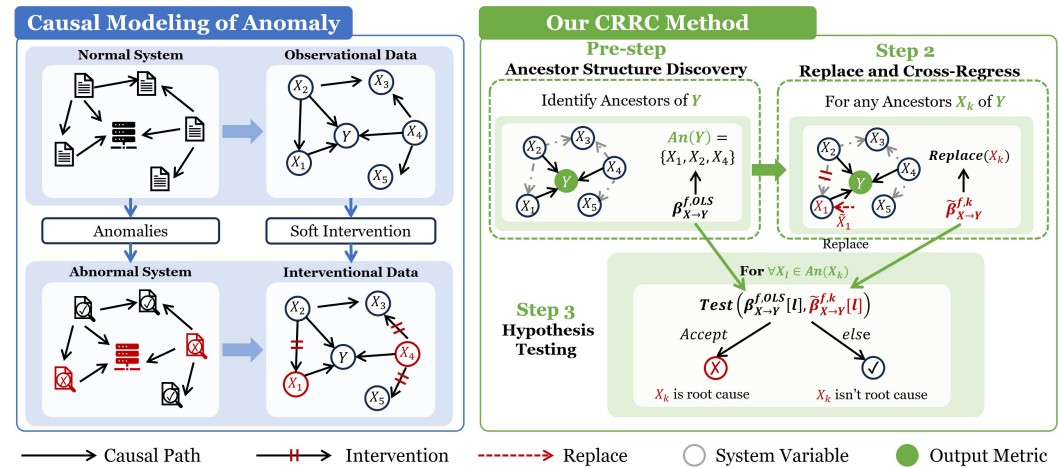

Figure 2: Illustration of our problem setup and proposed CRRC method.

**Lemma 1** (Ancestor Regression by Schultheiss & Bühlmann (2023))**.** *Considering the ordinary least squares (OLS) regression $f(Y)$ versus $X$ with a* **pre-defined non-linear** *function $f$, then each $An(Y)$ can be identified using the coupling regression coefficients of OLS regression:*

$$\beta_{X \to Y}^{f,OLS}[i] \neq 0 \Rightarrow X_i \in \mathrm{An}(Y), \tag{2}$$

*where $\beta_{X \to Y}^{f,OLS}[i]$ denotes the $i$-th coordinate of $\beta_{X \to Y}^{f,OLS}$.*

We note that **in our proposed framework, the pre-step of ancestor discovery is originally developed by Schultheiss & Bühlmann (2023).** Moreover, we note that any method capable of identifying ancestor relationships (Shimizu, 2014; Noè et al., 2019; Magliacane et al., 2016), can be directly plugged into our CRRC framework (see experimental details in Fig.5).

## 3.2 Cross-Regressing Intervened Ancestors

**Flaw of Causal-Discovery.** A naive strategy is to subsequently apply causal discovery on identified ancestors to find intervened variables, but it has limitations: (1) **Computational Challenges**. Methods like $\Phi$-PC (Ikram et al., 2022) must recover inter-connected edges in $X$, with redundant directions increasing cost in dense systems. (2) **High-dimensional Inefficiency**. Efficiency drops as $X$ grows: PC/FCI (Kocaoglu et al., 2019; Glymour et al., 2019) can scale exponentially with increasing variable size in $X$. (3) **Insufficient Granularity**. Markov equivalence is incomplete, i.e., often lacks the integrated causal graph needed for discovery-based root-cause analysis (Glymour et al., 2019).

**Our Solution.** To overcome the above-mentioned issues, we thus propose a **C**ross-**R**egressing-based **R**oot-**C**ause (CRRC) Method by comparing the regression coefficients of each variable $X_k \in An(Y)$ between the post and pre-intervention distributions with a replacement operation. In detail, our proposed CRRC (a) only concentrates on causal paths from $X \to Y$, thus avoiding the redundant computations; (b) does not recover the whole causal graph but to check the cross-regression statistics (feature-wise), thus exhibits superior identification efficiency:

---

1. **Original Regression.** Perform OLS regression $X \xrightarrow{OLS} f(Y)$ for some **non-linear** function $f$, obtaining the regression coefficient $\beta_{X \to Y}^{f,OLS}$.
2. **Cross Replacement.** For variable $X_k$, replacing the $k$-th column of the original feature matrix, i.e., $X_{.k}$, with the $k$-th column of the intervened $\tilde{X}_{.k}$.
3. **Cross Regression.** With the replaced data as $\tilde{X}^k$, performing OLS regression $\tilde{X}^k \xrightarrow{OLS} f(Y)$, and obtaining the corresponding regression coefficient $\tilde{\beta}_{X \to Y}^{f,k} \in \mathcal{R}^d$.
4. **Comparison.** Identifying whether $X_k \in X^{RC}$ by comparing at the $l$-th coordinate $\tilde{\beta}_{X \to Y}^{f,k}[l]$ with $\beta_{X \to Y}^{f,OLS}[l]$ obtained in equation 2 for $X_l \in An(X_k)$ .

---

**Remark 2.** *The ability of our method to identify intervened ancestors follows from three facts: (1) whether $X_k \in An(Y)$ is intervened is encoded in the SCM parameters of $X, Y$; (2) these parameters can be reformulated through linear structures; (3) in cross-regression, regression coefficients connect to these reformulated parameters (Proposition 1). Thus, variations in coefficients reveal changes in SCM parameters, indicating interventions on $X_k$.*

### 3.3 Algorithm Details

We then detail the whole process of CRRC by illustrating each step in Fig. 2 and in below.

**Pre-step: Ancestor Structure Discovery (ASD).** By recursively invoking Lemma 1, we can obtain the ancestor structure of $Y$, i.e., $ANS(Y) = \{An(X_k)\}$ for each $X_k \in An(Y)$, by alternatively performing OLS regression and ancestor identification (see detailed algorithm in Appendix D).

---

**Algorithm 1** Hypothesis Testing on Whether $\tilde{\beta}_{X \to Y}^{f,k}[l] \neq \beta_{X \to Y}^{f,OLS}[l]$.

---

**Input:** Estimated $\widehat{\tilde{\beta}_{X \to Y}^{f,k}}$ and $\widehat{\beta_{X \to Y}^{f,OLS}}$, with empirical variances of residuals $\widehat{\sigma^k}$ and $\widehat{\sigma}$, the cumulative distribution function (CDF) of normal distribution $\Psi$.

  1: Construct the null-hypothesis as $\tilde{\beta}_{X \to Y}^{f,k}[l] = \beta_{X \to Y}^{f,OLS}[l]$;

  2: Let $\kappa_l = \sqrt{n} \frac{|\widehat{\tilde{\beta}_{X \to Y}^{f,k}}[l] - \widehat{\beta_{X \to Y}^{f,OLS}}[l]|}{\widehat{\sigma^k} + \widehat{\sigma}}$ be empirical statistics, and Calculate $p_l = 2\left(1 - \Psi(\kappa_l)\right)$;

  3: Reject the null hypothesis If $p \leq 0.05$.

**Output:** Whether $\tilde{\beta}_{X \to Y}^{f,k}[l] \neq \beta_{X \to Y}^{f,OLS}[l]$.

---

**Algorithm 2** Procedure of our CRRC Framework.

---

**Input:** Observational and interventional empirical data $\mathcal{D} = x$ and $\tilde{\mathcal{D}} = \tilde{x}$, with normal and abnormal outcome variable as $y$ and $\tilde{y}$, and some non-linear function $f$.

  1: Record the original regression coefficient vector $\beta_{X \to Y}^{f,OLS}$.

  2: **Extracting Ancestor Structure**:

  3: Identify the ancestor structure (ANS) of $Y$, i.e., $ANS(Y)$, by pre-step.

  4: **Cross-regressing intervened Ancestors of $Y$**:

  5: **for** $X_k \in An(Y)$ **do**

  6:      Substitute corresponding column of $X_k$ in $x$ with the same column in $\tilde{x}$, and obtain $\tilde{X}_k$.

  7:      Perform $\tilde{X}^k \xrightarrow{OLS} f(Y)$ and obtain $\tilde{\beta}_{X \to Y}^{f,k}$.

  8:      **if** $\forall X_l \in An(X_k)$, $\tilde{\beta}_{X \to Y}^{f,k}[l] \neq \beta_{X \to Y}^{f,OLS}[l]$ **then** $X_k \in X^{RC}$ **else** $X_k \notin X^{RC}$.

**Output:** The root-cause variable set of $Y$ as $X^{RC}$.

---

**Main-Step: Cross-Regressing for Root-Cause Detection.** Now we justify whether each $X_k \in An(Y)$ is intervened, i.e., $X_k \in X^{RC}$. as outlined in lines 4-7, we perform cross-regression on $Y$ by replacing the column of each ancestor $X_k \in An(Y)$ in observational $X$ with the corresponding column in the interventional data matrix $\tilde{X}$. Given the regression coefficient vector $\tilde{\beta}_{X \to Y}^{f,k}$, we then compare $\tilde{\beta}_{X \to Y}^{f,k}$ with $\beta_{X \to Y}^{f,OLS}$ on each coordinate $l$ for $X_l \in An(X_k)$, and check whether the ancestor $X_k$ is intervened.

**Construction of Hypothesis Testing.** However, we also note that our criterion $\tilde{\beta}_{X \to Y}^{f,k} \neq \beta_{X \to Y}^{f,OLS}$ becomes non-informative in the finite-sample case, i.e., when instantiating our CRRC method. It is caused by the fact between regression coefficients with infinite samples and coefficients in the finite-sample case (Schultheiss & Bühlmann, 2023; Schultheiss et al., 2024). We further construct hypothesis testing to offer effective finite-sample criteria in Alg. 1, supported by Theorems 2 and 3.

## 4 Theory: CRRC Can Identify All Necessary Root-Causes

To show the Identifiability of our algorithm, we first have to introduce a definition on quantifying contributions of the exogenous noise variables (see detailed proofs of all theorems in Appendix E.):

**Definition 3** (Ancestor Weight Matrix). *The ancestor weight matrix $W$ with each entry $W_{ij}$ characterizing the contribution of $\Phi_j$ to $X_i$ in the SCM: $X = W\Phi$.*

In similar, one can write interventional random vectors as $\tilde{X} = \tilde{W}\Phi$, where we use $\tilde{W}$ to denote the intervened version of $W$. For substituted data $\tilde{x}^k$, we write the corresponding random vector $\tilde{X}^k = \tilde{W}^k \Phi$, where $\tilde{W}^k$ keeps the same with $W$ except for the $k$-th row. Based on Def. 3, we then derive our first proposition:

**Proposition 1.** *Assuming the cross-interventional moment matrix $\mathbb{E}[\tilde{X}^k \left( \tilde{X}^k \right)^T]$ is invertible for each $k \in [p]$, and the observational matrix $\mathbb{E}[XX^T]$ is invertible, the following equation holds:*

$$(\tilde{W}^k)^T \tilde{\beta}_{X \to Y}^{f,k} = W^T \beta_{X \to Y}^{f,OLS}. \tag{3}$$

To ensure the identification, we then introduces another regularization condition:

**Assumption 1** (Regularity on Ancestor Weight Matrix). *For each $i \in [p]$, once $X_i$ is intervened, $W_{ij}$ will vary for each $j \in An(i)$.*

The above regularity condition states that the intervention on some variable $X_i$ will cause changes in effects from its ancestors. By contrast, the soft-intervention does not hold for $X_i$ if nearly every ancestor's effect on $X_i$ does not vary. Finally, based on Proposition 1 and Assumption 1, we present the **key identification** theorem in the asymptotic regime in below:

**Theorem 1** (Identifiability on Necessary Root-Causes). *For any $k \in [p]$ and $X_k \in An(Y)$, $X_k \in X^{RC}$ if and only if $\tilde{\beta}_{X \to Y}^{f,k}[l] \neq \beta_{X \to Y}^{f,OLS}[l]$ for each $X_l \in An(X_k)$.*

Based on Def. 2, Theorem 1 informs that comparing regression coefficients $\tilde{\beta}_{X \to Y}^{f,k}[l]$ and $\beta_{X \to Y}^{f,OLS}[l]$ for any $X_l \in An(X_k)$ is sufficient to check that whether $X_k$ itself is an outcome-oriented root-cause.

## 4.1 ASYMPTOTIC ANALYSIS: MOTIVATION OF HYPOTHESIS TESTING

We then check the validity of our hypothesis testing proposed in Alg. 1. In below, we first prove that the difference between regression coefficients $\widehat{\tilde{\beta}_{X \to Y}^{f,k}}[l]$ and $\widehat{\beta_{X \to Y}^{f,OLS}}[l]$ estimated using finite samples converges to a zero-meaned Gaussian distribution:

**Theorem 2** (Asymptotic Gaussianity). *Assume that $E\left\{ f\left(Y\right)^2 \right\} < \infty$, $E\left(X_k^4\right) < \infty$ for all $k$, and $\tilde{\beta}_{X \to Y}^{f,k}$ and $\beta_{X \to Y}^{f,OLS}$ exists. Then for variables in $An(X_k)$ with $\tilde{\beta}_{X \to Y}^{f,k}[l] = \beta_{X \to Y}^{f,OLS}[l]$, the following convergence in distribution holds:*

$$\sqrt{n} \left( \widehat{\tilde{\beta}_{X \to Y}^{f,k}}[l] - \widehat{\beta_{X \to Y}^{f,OLS}}[l] \right) \xrightarrow{\mathbb{D}} \mathcal{N} \left( 0, \mathbb{E}\left[\mathcal{E}^2\right] + \mathbb{E}\left[(\mathcal{E}^k)^2\right] \right), \tag{4}$$

*where $\mathcal{E}$ and $\mathcal{E}^k$ refers to expected residual error of the regressions, respectively.*

Furthermore, we estimate the variance of the Gaussian in finite-sample regime such that one can construct hypothesis:

**Theorem 3** (Variance Estimation). *The following convergence in probability holds:*

$$(\hat{\epsilon})^2 + \left(\hat{\epsilon}^k\right)^2 \xrightarrow{\mathbb{P}} \mathbb{E}\left[\mathcal{E}^2\right] + \mathbb{E}\left[(\mathcal{E}^k)^2\right], \tag{5}$$

*where $(\hat{\epsilon})^2 := \frac{\|f(y) - x\widehat{\beta_{X \to Y}^{f,OLS}}\|_2^2}{n-p}$ and $\left(\hat{\epsilon}^k\right)^2 := \frac{\|f(y) - x^k \widehat{\tilde{\beta}_{X \to Y}^{f,k}}\|_2^2}{n-p}$.*

## 4.2 COMPUTATIONAL COMPLEXITY ANALYSIS

We finally analyze the polynomial complexity of our proposed cross-regression step w.r.t. our proposed cross-regression component. It is obvious that the maximum number to perform OLS regression across observational and interventional data is $\mathcal{O}(p)$. Hence, the upper bound of the cross-regression stage is $\mathcal{O}(np^3)$, due to the intrinsic complexity of OLS regression as $\mathcal{O}(np^2)$. In summary, the overall computational complexity is $\mathcal{O}(np^3)$.

Table 1: Results (Precision or Recall@k) on synthetic data, including the Erdős-Rényi (ER) and the Scale-Free (SF) Graph (Top-1 metric is not considered as $X^{RC}$ contains more than 1 root-causes).

| Dataset | | | Erdős-Rényi Graph | | | | | Scale-Free Graph | | | | |
|---|---|---|---|---|---|---|---|---|---|---|---|---|
| # Top-K | Type | Algo | N=20 | N=40 | N=60 | N=80 | N=100 | N=20 | N=40 | N=60 | N=80 | N=100 |
| k=3 | COA | Eplison | $0.00_{\pm0.00}$ | $0.00_{\pm0.00}$ | $0.00_{\pm0.00}$ | $0.00_{\pm0.00}$ | $0.27_{\pm0.13}$ | $0.07_{\pm0.13}$ | $0.07_{\pm0.13}$ | $0.00_{\pm0.00}$ | $0.13_{\pm0.16}$ | $0.13_{\pm0.16}$ |
| | | RW | $0.23_{\pm0.00}$ | $0.13_{\pm0.00}$ | $0.12_{\pm0.00}$ | $0.00_{\pm0.00}$ | $0.40_{\pm0.25}$ | $0.40_{\pm0.25}$ | $0.40_{\pm0.25}$ | $0.40_{\pm0.25}$ | $0.40_{\pm0.25}$ | $0.13_{\pm0.16}$ |
| | DB-RCA | Traversal | $0.33_{\pm0.00}$ | $0.33_{\pm0.00}$ | $0.00_{\pm0.00}$ | $0.00_{\pm0.00}$ | $0.00_{\pm0.00}$ | $0.00_{\pm0.00}$ | $0.00_{\pm0.00}$ | $0.00_{\pm0.00}$ | $0.00_{\pm0.00}$ | $0.00_{\pm0.00}$ |
| | | CIRCA | $0.34_{\pm0.13}$ | $0.42_{\pm0.13}$ | $0.33_{\pm0.00}$ | $0.30_{\pm0.13}$ | $0.16_{\pm0.00}$ | $0.13_{\pm0.16}$ | $0.07_{\pm0.13}$ | $0.07_{\pm0.13}$ | $0.00_{\pm0.00}$ | $0.00_{\pm0.00}$ |
| | | RCD | $0.19_{\pm0.05}$ | $0.11_{\pm0.07}$ | $0.07_{\pm0.01}$ | $0.15_{\pm0.09}$ | $0.08_{\pm0.05}$ | $0.06_{\pm0.02}$ | $0.12_{\pm0.10}$ | $0.04_{\pm0.05}$ | $0.10_{\pm0.08}$ | $0.16_{\pm0.09}$ |
| | | DeepITE | $0.49_{\pm0.01}$ | $0.60_{\pm0.02}$ | $0.52_{\pm0.02}$ | $0.61_{\pm0.01}$ | $0.57_{\pm0.01}$ | $0.64_{\pm0.02}$ | $0.64_{\pm0.02}$ | $0.57_{\pm0.01}$ | $0.64_{\pm0.02}$ | $0.64_{\pm0.02}$ |
| | | UT-IGSP | $0.16_{\pm0.01}$ | $0.08_{\pm0.00}$ | $0.05_{\pm0.00}$ | $0.04_{\pm0.00}$ | $0.03_{\pm0.00}$ | $0.14_{\pm0.00}$ | $0.07_{\pm0.00}$ | $0.05_{\pm0.00}$ | $0.04_{\pm0.00}$ | $0.03_{\pm0.00}$ |
| | CF-RCA | CF-Attr | $0.17_{\pm0.10}$ | $0.18_{\pm0.08}$ | $0.21_{\pm0.09}$ | $0.22_{\pm0.09}$ | $0.20_{\pm0.07}$ | $0.28_{\pm0.06}$ | $0.11_{\pm0.06}$ | $0.29_{\pm0.06}$ | $0.13_{\pm0.10}$ | $0.21_{\pm0.05}$ |
| | | TOCA | $0.30_{\pm0.08}$ | $0.24_{\pm0.06}$ | $0.26_{\pm0.08}$ | $0.27_{\pm0.07}$ | $0.24_{\pm0.09}$ | $0.11_{\pm0.06}$ | $0.22_{\pm0.06}$ | $0.20_{\pm0.10}$ | $0.16_{\pm0.06}$ | $0.21_{\pm0.06}$ |
| | | IDI | $0.13_{\pm0.07}$ | $0.19_{\pm0.06}$ | $0.12_{\pm0.10}$ | $0.14_{\pm0.08}$ | $0.29_{\pm0.07}$ | $0.28_{\pm0.08}$ | $0.22_{\pm0.07}$ | $0.26_{\pm0.05}$ | $0.15_{\pm0.10}$ | $0.16_{\pm0.06}$ |
| | ST-RCA | LinearEST | $0.39_{\pm0.09}$ | $0.45_{\pm0.07}$ | $0.37_{\pm0.07}$ | $0.41_{\pm0.03}$ | $0.42_{\pm0.05}$ | $0.44_{\pm0.03}$ | $0.49_{\pm0.03}$ | $0.47_{\pm0.04}$ | $0.49_{\pm0.03}$ | $0.44_{\pm0.03}$ |
| | | iSCAN | $0.35_{\pm0.03}$ | $0.23_{\pm0.05}$ | $0.14_{\pm0.09}$ | $0.02_{\pm0.03}$ | $0.02_{\pm0.02}$ | $0.17_{\pm0.03}$ | $0.18_{\pm0.15}$ | $0.23_{\pm0.20}$ | $0.05_{\pm0.06}$ | $0.02_{\pm0.03}$ |
| | | **CRRC (Ours)** | $0.93_{\pm0.07}$ | $0.93_{\pm0.03}$ | $0.90_{\pm0.03}$ | $0.93_{\pm0.03}$ | $0.73_{\pm0.05}$ | $1.00_{\pm0.00}$ | $1.00_{\pm0.00}$ | $1.00_{\pm0.00}$ | $1.00_{\pm0.00}$ | $1.00_{\pm0.00}$ |
| k=5 | COA | Eplison | $0.06_{\pm0.00}$ | $0.06_{\pm0.00}$ | $0.06_{\pm0.00}$ | $0.00_{\pm0.00}$ | $0.06_{\pm0.00}$ | $0.27_{\pm0.13}$ | $0.07_{\pm0.13}$ | $0.07_{\pm0.13}$ | $0.00_{\pm0.00}$ | $0.13_{\pm0.16}$ |
| | | RW | $0.00_{\pm0.00}$ | $0.00_{\pm0.00}$ | $0.00_{\pm0.00}$ | $0.00_{\pm0.00}$ | $0.00_{\pm0.00}$ | $0.67_{\pm0.21}$ | $0.67_{\pm0.21}$ | $0.67_{\pm0.21}$ | $0.67_{\pm0.21}$ | $0.67_{\pm0.21}$ |
| | DB-RCA | Traversal | $0.33_{\pm0.00}$ | $0.33_{\pm0.00}$ | $0.00_{\pm0.00}$ | $0.00_{\pm0.00}$ | $0.00_{\pm0.00}$ | $0.00_{\pm0.00}$ | $0.00_{\pm0.00}$ | $0.00_{\pm0.00}$ | $0.00_{\pm0.00}$ | $0.00_{\pm0.00}$ |
| | | CIRCA | $0.20_{\pm0.27}$ | $0.27_{\pm0.13}$ | $0.27_{\pm0.13}$ | $0.00_{\pm0.00}$ | $0.07_{\pm0.13}$ | $0.13_{\pm0.16}$ | $0.07_{\pm0.13}$ | $0.07_{\pm0.13}$ | $0.00_{\pm0.00}$ | $0.00_{\pm0.00}$ |
| | | RCD | $0.19_{\pm0.05}$ | $0.14_{\pm0.07}$ | $0.14_{\pm0.09}$ | $0.12_{\pm0.08}$ | $0.05_{\pm0.05}$ | $0.07_{\pm0.05}$ | $0.06_{\pm0.04}$ | $0.14_{\pm0.04}$ | $0.07_{\pm0.03}$ | $0.08_{\pm0.09}$ |
| | | DeepITE | $0.54_{\pm0.02}$ | $0.55_{\pm0.02}$ | $0.52_{\pm0.01}$ | $0.61_{\pm0.01}$ | $0.52_{\pm0.01}$ | $0.64_{\pm0.01}$ | $0.59_{\pm0.02}$ | $0.57_{\pm0.02}$ | $0.69_{\pm0.01}$ | $0.64_{\pm0.01}$ |
| | | UT-IGSP | $0.16_{\pm0.01}$ | $0.08_{\pm0.00}$ | $0.05_{\pm0.00}$ | $0.04_{\pm0.00}$ | $0.03_{\pm0.00}$ | $0.14_{\pm0.00}$ | $0.07_{\pm0.00}$ | $0.05_{\pm0.00}$ | $0.04_{\pm0.00}$ | $0.03_{\pm0.00}$ |
| | CF-RCA | CF-Attr | $0.21_{\pm0.06}$ | $0.18_{\pm0.10}$ | $0.12_{\pm0.06}$ | $0.10_{\pm0.07}$ | $0.14_{\pm0.06}$ | $0.11_{\pm0.06}$ | $0.24_{\pm0.09}$ | $0.17_{\pm0.07}$ | $0.24_{\pm0.09}$ | $0.14_{\pm0.09}$ |
| | | TOCA | $0.21_{\pm0.07}$ | $0.26_{\pm0.06}$ | $0.18_{\pm0.08}$ | $0.25_{\pm0.09}$ | $0.20_{\pm0.10}$ | $0.21_{\pm0.07}$ | $0.11_{\pm0.09}$ | $0.11_{\pm0.08}$ | $0.20_{\pm0.09}$ | $0.25_{\pm0.07}$ |
| | | IDI | $0.26_{\pm0.08}$ | $0.19_{\pm0.05}$ | $0.24_{\pm0.08}$ | $0.19_{\pm0.05}$ | $0.28_{\pm0.07}$ | $0.29_{\pm0.10}$ | $0.12_{\pm0.06}$ | $0.11_{\pm0.05}$ | $0.28_{\pm0.05}$ | $0.16_{\pm0.06}$ |
| | ST-RCA | LinearEST | $0.39_{\pm0.09}$ | $0.45_{\pm0.07}$ | $0.37_{\pm0.07}$ | $0.41_{\pm0.03}$ | $0.42_{\pm0.05}$ | $0.44_{\pm0.03}$ | $0.49_{\pm0.03}$ | $0.47_{\pm0.04}$ | $0.49_{\pm0.03}$ | $0.44_{\pm0.03}$ |
| | | iSCAN | $0.35_{\pm0.03}$ | $0.23_{\pm0.05}$ | $0.14_{\pm0.09}$ | $0.02_{\pm0.03}$ | $0.02_{\pm0.02}$ | $0.17_{\pm0.16}$ | $0.18_{\pm0.15}$ | $0.23_{\pm0.20}$ | $0.05_{\pm0.06}$ | $0.02_{\pm0.03}$ |
| | | **CRRC (Ours)** | $0.93_{\pm0.07}$ | $0.93_{\pm0.03}$ | $0.90_{\pm0.03}$ | $0.93_{\pm0.03}$ | $0.73_{\pm0.05}$ | $1.00_{\pm0.00}$ | $1.00_{\pm0.00}$ | $1.00_{\pm0.00}$ | $1.00_{\pm0.00}$ | $1.00_{\pm0.00}$ |

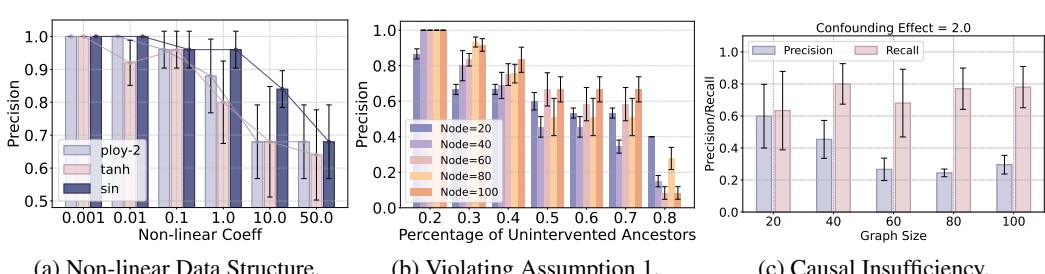

(a) Non-linear Data Structure.     (b) Violating Assumption 1.     (c) Causal Insufficiency.

Figure 3: Robustness Testing of our proposed CRRC method against violations of assumptions.

# 5 EXPERIMENTS

**Baselines.** We position our CRRC against 13 state-of-the-art (SOTA) methods, spanning four areas of relevance: Correlation-based Anomalies (COA), Discover-based RCA (DB-RCA), Counterfactual-based RCA (CF-RCA), and Statistics-based RCA (ST-RCA), as follows (see details in Appendix G.1): **COA.** We select the $\epsilon$-diagnosis (Shan et al., 2019) and random_walk (Yu et al., 2021); **DB-RCA.** We select 5 methods, encompassing the traversal method (Chen et al., 2014b), UT-IGSP (Wang et al., 2017), RCD (Ikram et al., 2022), DeepITE Tao et al. (2024), and CIRCA (Li et al., 2022); **CF-RCA.** We select 3 methods, encompassing the CF-attr methods using Shapley-based outlier score (Budhathoki et al., 2022), TOCA (Okati et al., 2024) the relaxed version of CF-attr, and the recent proposed in-distribution method IDI (anonymous, 2025). **ST-RCA.** We pick the LinearEST method (Varici et al., 2021) and non-linear iSCAN method (Chen et al., 2024) (see Tab. 5 in the Appendix). We set $f(Y) = Y^3$ throughout our experiments (Schultheiss & Bühlmann, 2023).

**Benchmarks and Metrics.** We evaluate on three benchmarks: fully synthetic data, semi-synthetic supply chain data (Budhathoki et al., 2022), and two real-world datasets, including the Sachs protein data (Sachs et al., 2005) and the Petshop benchmark (Hardt et al., 2024) (see details in Appendix G.2). Synthetic data are generated from two random DAGs: Erdős-Rényi (ER) and Scale-free (SF). For evaluation, we adopt precision and Recall@K, accommodating varied identification outputs (see details in Tab. 5 with Appendix G.1). Results are averaged over 10 trials with error bars denoting standard deviations (except for Petshop data with only averaged reports (Nagalapatti et al., 2025)).

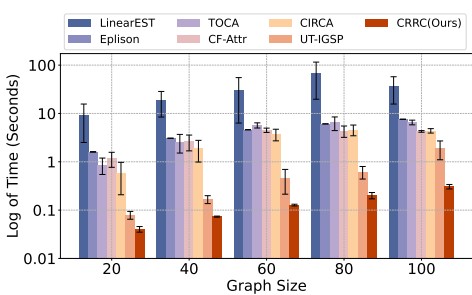

Figure 4: The running efficiency analysis of each baselines (inference time per sample).

Table 2: Results on the semi-synthetic data, including the Erdős-Rényi (ER) Graph and the Scale-Free (SF) Graph.

| Type | Algo | Semi-Synthetic Supply | | | Real-Sachs | |
|---|---|---|---|---|---|---|
| | | k=2↑ | k=3↑ | k=5↑ | k=1↑ | k=3↑ |
| COA | Eplison | $0.05_{\pm0.03}$ | $0.08_{\pm0.10}$ | $0.06_{\pm0.09}$ | $0.08_{\pm0.03}$ | $0.16_{\pm0.05}$ |
| | RW | $0.50_{\pm0.00}$ | $0.50_{\pm0.00}$ | $0.00_{\pm0.00}$ | $0.06_{\pm0.04}$ | $0.16_{\pm0.04}$ |
| DB-RCA | Traversal | $0.09_{\pm0.01}$ | $0.10_{\pm0.02}$ | $0.11_{\pm0.09}$ | $0.18_{\pm0.08}$ | $0.05_{\pm0.04}$ |
| | CIRCA | $0.20_{\pm0.25}$ | $0.30_{\pm0.25}$ | $0.30_{\pm0.25}$ | $0.21_{\pm0.00}$ | $0.20_{\pm0.00}$ |
| | RCD | $0.15_{\pm0.10}$ | $0.19_{\pm0.10}$ | $0.12_{\pm0.07}$ | $0.21_{\pm0.00}$ | $0.29_{\pm0.00}$ |
| | DeepITE | $0.18_{\pm0.00}$ | $0.24_{\pm0.18}$ | $0.50_{\pm0.12}$ | $0.65_{\pm0.00}$ | $0.71_{\pm0.00}$ |
| | UT-IGSP | $0.19_{\pm0.01}$ | $0.19_{\pm0.01}$ | $0.19_{\pm0.01}$ | $0.58_{\pm0.02}$ | $0.80_{\pm0.01}$ |
| CF-RCA | CF-Attr | $0.09_{\pm0.01}$ | $0.13_{\pm0.07}$ | $0.10_{\pm0.08}$ | $0.01_{\pm0.00}$ | $0.00_{\pm0.00}$ |
| | TOCA | $0.10_{\pm0.06}$ | $0.08_{\pm0.01}$ | $0.12_{\pm0.02}$ | $0.07_{\pm0.07}$ | $0.16_{\pm0.06}$ |
| | IDI | $0.10_{\pm0.10}$ | $0.10_{\pm0.06}$ | $0.15_{\pm0.07}$ | $0.17_{\pm0.09}$ | $0.08_{\pm0.03}$ |
| ST-RCA | LinearEST | $0.40_{\pm0.00}$ | $0.40_{\pm0.00}$ | $0.40_{\pm0.00}$ | $0.45_{\pm0.19}$ | $0.47_{\pm0.15}$ |
| | iSCAN | $0.11_{\pm0.09}$ | $0.11_{\pm0.09}$ | $0.11_{\pm0.09}$ | $0.11_{\pm0.09}$ | $0.11_{\pm0.09}$ |
| | CRRC (Ours) | $1.00_{\pm0.00}$ | $1.00_{\pm0.00}$ | $1.00_{\pm0.00}$ | $0.75_{\pm0.00}$ | $1.00_{\pm0.00}$ |

**Questions to Investigate.** Our experiments address three questions: (1) Can CRRC accurately localize root causes across varying graph and sample sizes? (2) How does CRRC compare with baselines that intersect INI results with ancestor discovery (Schultheiss & Bühlmann, 2023)? (3) How robust is CRRC under assumption violations on synthetic Erdős-Rényi data? **Notably, CRRC requires no dependency or causal graph, consistent with the OORCA setting where no structural knowledge is assumed.**

**Performance Analysis.** Analysis of Tab. 1 and 2 yields three insights: (1) Traditional COA Methods Fail. They miss root causes in nearly all settings, underscoring the need for causality. (2) INI Methods Fall Short. Both discovery- and statistics-based INI capture redundant causes (e.g., LinearEST, UT-ISGP with low precision), highlighting the necessity of separating RCA from INI. (3) CRRC Excels. Our method consistently outperforms baselines across top-K metrics, benefiting from cross-regression with ancestor search (Schultheiss & Bühlmann, 2023).

**Running-Time Analysis.** We analyze runtime by varying causal graph size $p \in [20, 40, 60, 80, 100]$. For each $p$, 10 trials are conducted on 2000 samples, reporting average total time (training + inference). As shown in Fig. 4, CRRC achieves the lowest runtime across graph sizes, demonstrating the efficiency of cross-regression.

**Non-Linear Data Structure.** We perform empirical analysis by introducing non-linearity into the SCM, violating our linear setup (synthetic data with graph size as 40). We add three types of non-linear function to generate each $X_i$, and tune the non-linear coefficient (see details in Appendix G.4). As shown in Fig. 3a, our proposed CRRC exhibits robustness against non-linearity.

**Robustness Towards Violation of Assumption 1.** To verify the robustness of our method against the key Assumption 1, we tune the ratio of varied $W_{ji}$ within the whole $|An(X_i)|$ for each intervened $X_i$ (see details in Appendix G.3). As shown in Fig 3b, our CRRC exhibits enough robustness when more than half of $An(X_i)$ intervened, and poor performance occurs when nearly all the $W_{ji}$ of $An(X_i)$ will not intervened, while such cases imply the non-existence of intervention on $X_i$.

**Performance With Causal Insufficiency.** We test the performance of our methods by violating the causal sufficiency assumption (synthetic data with confounding effect as 2.0, see Details in Appendix G.6). We observe that the number of unmeasured confounders plays a key role in degrading the precision metric with redundancy. Surprisingly, we find that our CRRC maintains high recall metric, implying that most RCs are still identified.

**Other Robustness Test.** In addition, we verify that our method exhibits robustness in diverse imperfect data environment: (1) Varying size of the number of underlying set of $X^{RC}$ (see Appendix G.8); (2) The sample efficiency on anomalous data (see Appendix G.9); (3) Varying the connection density of the causal graph (see Appendix G.10).

**Case Study: Effect of Ancestor Discovery.** We further examine two questions: (a) what if OORCA is implemented as *the intersection between existing RCA baselines and ancestors mined by Schultheiss & Bühlmann (2023)*? (b) how does CRRC perform with *alternative ancestor-mining strategies* (Magliacane et al., 2016; Shimizu, 2014)? For (a), we construct four baselines by running competitive RCA methods and ancestor regression in parallel, then intersecting INI results with the ancestor set. As shown in Fig. 5, CRRC surpasses these brute-force baselines by a wide margin. For (b), we implement CRRC with ACI and LiNGAM, and results confirm its adaptability to different

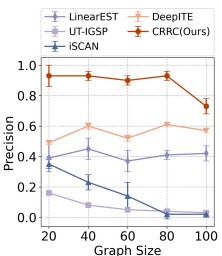 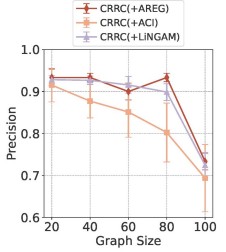

Figure 5: **Left.** Comparison with different OORCA baselines as intersection between ancestor discovery and intervention identification approaches; **Right.** Comparison across different ancestor discovery baselines, i.e., AREG, ACI and LiNGAM.

Table 3: Results (Recall@k) on a real-world, outcome-oriented Petshop benchmark.

| Type | Algo | Low | | High | | Temporal | |
|---|---|---|---|---|---|---|---|
| | | k=1 ↑ | k=3 ↑ | k=1 ↑ | k=3 ↑ | k=1 ↑ | k=3 ↑ |
| COA | RW | 0.00 | 0.10 | 0.00 | 0.20 | 0.00 | 0.33 |
| | Epilson | 0.00 | 0.00 | 0.00 | 0.00 | 0.17 | 0.17 |
| DB-RCA | CIRCA | 0.40 | 0.60 | 0.30 | 0.40 | 0.67 | 1.00 |
| | Traversal | 0.30 | 0.50 | 0.20 | 0.30 | 1.00 | 1.00 |
| | RCD | 0.21 | 0.75 | 0.07 | 0.00 | 0.75 | 0.75 |
| | DeepITE | 0.65 | 0.75 | 0.50 | 0.65 | 0.90 | 0.95 |
| CF-RCA | TOCA | 0.40 | 0.40 | 0.20 | 0.20 | 0.00 | 0.00 |
| | CF-Attr | 0.40 | 0.60 | 0.40 | 0.70 | 0.50 | 0.50 |
| | IDI | 0.80 | 0.85 | 0.65 | 0.70 | 1.00 | 1.00 |
| ST-RCA | LinearEST | 0.50 | 0.50 | 0.30 | 0.60 | 0.50 | 0.60 |
| | iSCAN | 0.60 | 0.65 | 0.50 | 0.55 | 0.80 | 0.90 |
| Ours | CRRC | 0.85 | 0.90 | 0.80 | 0.90 | 1.00 | 1.00 |

ancestor-mining strategies, demonstrating CRRC as a flexible, general framework for outcome-oriented root-cause identification.

## 6 RELATED WORK

In this section, we briefly review literature on RCA, which can be divided into discovery-based RCA, i.e., causal discovery-based RCA (Li et al., 2022; Ikram et al., 2022; Zheng et al., 2024; Yang et al., 2024), counterfactual reasoning-based RCA (Budhathoki et al., 2022; Okati et al., 2024; Nguyen et al., 2024), and statistical RCA (Varici et al., 2021; 2022; Chen et al., 2014b) (see more detailed related work in Appendix B).

**Discovery-Based RCA**. These methods build on causal discovery (Glymour et al., 2019), typically following a two-stage paradigm: first recover the causal graph (e.g., PC (Ikram et al., 2022), DAG-GNN (Zheng et al., 2024)), then apply retrieval algorithms such as random walks (Ma et al., 2019; 2020a; Wang et al., 2018; Ma et al., 2020b), PageRank (Kim et al., 2013; Xin et al., 2023), or depth search (Chen et al., 2014a; Liu et al., 2021; Guan et al., 2019) to identify root causes. Recent extensions leverage causal discovery on mixed data (Jaber et al., 2020), inspiring methods like $\Phi$-PC (Ikram et al., 2022) and others (Li et al., 2022; Yang et al., 2024).

**Counterfactual Reasoning-Based RCA** methods model the appear of anomalies as so-called ""structure-preserving interventions" (Budhathoki et al., 2022; Okati et al., 2024; Nguyen et al., 2024; Nagalapatti et al., 2025). To be specific, the CF-Attr method (Budhathoki et al., 2022) combines the Shapley-value with Z-score metric to quantify each variable's contribution to the outlier, and Nguyen et al. (2024) extends such framework to the case that the edges of the causal graph are also intervened. In recent, Nagalapatti et al. (2025) extends such counterfactual-reasoning framework into the out-of-distribution domain, exhibiting robust performance.

**Statistical RCA.** Statistical RCA checks whether some **pre-constructed** statistics exhibit shift across normal and abnormal data (Varici et al., 2021; 2022; Chen et al., 2024), e.g., entries in the precision matrices (Varici et al., 2021) or score functions (Chen et al., 2024).

## 7 CONCLUSION

This paper studies a novel and challenging problem named Outcome-Oriented Root-Cause Analysis (OORCA) by identifying all necessary root-causes. By regressing across normal and abnormal data and comparing regression coefficients, we propose a cross-regression strategy to identify intervened variables affecting the outcome variable. In future work, we focus on the extension of our work to two more challenging settings: (a) RCA in causal insufficient systems with hidden confounders, where additional observations, i.e., instrumental variable (Baiocchi et al., 2014), is required to accurate identification; (b) RCA in non-linear data structure. Although some non-linear robustness is exhibited empirically, we plan to extend our CRRC by regression in kernel spaces, with theoretically guaranteed non-linear RCA capability.

ETHICS STATEMENT

This paper proposes a novel root-cause identification method, tailored for the outcome-oriented root-cause analysis problem. Potential broader impacts include extending our proposed method for localizing and fixing anomalies towards realistic applications with large, running systems. In sometime, anomalies without localizing and fixing operations in time might raise risk in both the efficiency and safeguard considerations. We hope that our proposed CRRC framework can aid large, running systems reduce such risk.

REPRODUCIBILITY STATEMENT

We provide detailed descriptions of our framework, theoretical results, and experimental settings in the paper and appendix. All datasets used are publicly available, and the current description of our method is sufficient for full reproducibility. If the paper is accepted, we will be glad to release the complete implementation to further support the research community.

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

APPENDIX

We provide supplementary documents to support our research. The details of Large Language Model usage are presented in Section A. We provide more thorough review on related work in Section B with detailed notations in Section C. Besides, we put detailed algorithmic procedures in Section D. Consequently, we leave the proof details with complexity analysis in Section E and F, respectively. Finally, we put experimental details, including the baseline setups, data preparations, with more experimental results in Section G.

## A  LARGE LANGUAGE MODEL USAGE

In this paper, we clarify that large language models (LLMs) are employed solely to support and refine the writing process. Specifically, we use LLMs to provide sentence-level suggestions and to enhance the overall fluency of the text.

## B  MORE RELATED WORK

We supplement more closed-connected related work here. As intervention identification (INI) also aims to identify interventions given observational and interventional data, we first introduce the area of INI as a closely-related area (Yang et al., 2024) in below:

**Causal Discovery-Based INI.** INI aims to recover intervened targes (IT) from the combination of both observational and inverventional data Wang et al. (2017). One branch of INI approaches couples the estimation of intervention targets together with the identification of the underlying causal graphs (Wang et al., 2017; Jaber et al., 2020; Ikram et al., 2022). With causal sufficient systems, the UT-IGSP (Wang et al., 2017) method first explores the interventional Markov equivalence class (I-MEC) from mixed data through permutation searches. In the case of causal insufficiency, Jaber et al. (2020) characterizes the $\Psi$-Markov class from mixed data. Meanwhile, JCI (Mooij et al., 2020) leverages context variables for data pooling. In recent, DeepITE (Tao et al., 2024) designs a deep variational causal graph autoencoder to harnesses correlated information for INI.

**Statistics-Based INI.** As discovery-based INI approaches are vulnerable to mis-specification of identified causal graphs, the other INI branch directly constructs statistics reflecting the intervention operations within a given causal framework (Varici et al., 2021; 2022; Chen et al., 2024). In linear causal models, LinearEST (Varici et al., 2021; 2022) identifies ITs by constructing the variation of precision matrices, which avoids the search of causal graphs. In non-linear cases, LIT (Yang et al., 2024) utilizes non-linear Independent Component Analysis to recognize ITs across multiple environments, and iSCAN (Chen et al., 2024) proposes to recover ITs by constructing and examining the variances of scores before and after interventions. However, the scalability of the above methods falls in short with increasing graph size. In this paper, we contribute a novel but efficient statistics based on regression coefficients.

## C  NOTATIONS DETAILS

We present detailed summary of notations we used in problem setup, algorithm design and theoretical analysis in Tab 4.

## D  DETAILED ALGORITHM OF PRE-STEP ANCESTOR DISCOVERY

We then detail the algorithm of Ancestor Structure Discovery (ASD) in below:

Table 4: Summary of notions with their definitions.

| Notation | Definition |
|---|---|
| $X$ | Normal (observational) System Variables. |
| $Y$ | Normal (observational) outcome variable. |
| $\tilde{X}$ | Abnormal (interventional) System Variables. |
| $\tilde{Y}$ | Abnormal (interventional) outcome variable. |
| $p$ | Number of variables in the system. |
| $G, \mathcal{G}$ | Linear Matrix and Causal Graph of the SCM. |
| $\Phi$ | Exogenous Variables. |
| $\epsilon$ | Upper-bounded Variance of $\Phi$. |
| $Pa(X^*), Ch(X^*), An(X^*)$ and $De(X^*)$ | The set of parents, children, ancestors and descendant of variable $X^*$. |
| $X_{.k}$ and $\tilde{X}_{.k}$ | $k$-th column of observational and interventional data. |
| $\tilde{x}^k$ and $\tilde{X}^k$ | Substituted Data in the column k. |
| $X^{RC}$ | The set of root-causes. |
| $\beta_{X_i \to Y}$ | Ancestor Effect of variable $X_i$ to $Y$. |
| $\beta_{X \to Y}^{f,OLS}, \beta_{X \to Y}^{f,OLS}[l]$ | OLS regression coefficient of $X$ on $f(Y)$ and its $l$-th coordinate. |
| $\tilde{X}^k \xrightarrow{OLS} f(Y), \tilde{X}^k \xrightarrow{OLS} f(Y)[l]$ | OLS regression coefficient of $\tilde{X}^k$ on $f(Y)$ and its $l$-th coordinate. |
| $ANS(Y)$ | The two-layer ancestor set of $Y$. |
| $\widehat{\tilde{\beta}_{X \to Y}^{f,k}}$ and $\widehat{\beta_{X \to Y}^{f,OLS}}$ | Empirical estimated OLS regression coefficients of $\beta_{X \to Y}^{f,OLS}$ and $\tilde{X}^k \xrightarrow{OLS} f(Y)$. |
| $\Psi$ | The CDF of normal distribution. |
| $n$ | Number of samples. |
| $W$ | Ancestor weight matrix in Def 3. |
| $\tilde{W}^k$ | Substituted version of $W$ on the $k$-th column. |
| $\epsilon, \hat{\epsilon}, \hat{\epsilon}^k$ | Residual and ite empirical estimations. |

**Algorithm 3** Ancestor Structure Discovery (ASD).

**Input:** Observational and interventional empirical data $\mathcal{D} = x$ and $\tilde{\mathcal{D}} = \tilde{x}$, with normal and abnormal outcome variable as $y$ and $\tilde{y}$ and some non-linear function $f$.

1: Perform $X \xrightarrow{OLS} f(Y)$;
2: Using Lemma 1 to identify $An(Y)$;
3: **for** $X_k \in An(Y)$ **do**
4:     Perform $X \xrightarrow{OLS} f(X_k)$ and identify $An(X_k)$ with Lemma 1;

**Output:** The ancestor structure as $ANS(Y) = \{An(X_k)\}$ for each $X_k \in An(Y)$.

# E    THEORETICAL PROOF ON THE SOUNDNESS

In below, following previous protocols Schultheiss et al. (2024), we introduce additional notations for the proofs present later in this section. We use the subindex $-k$ to denote all variables in except for $k$, e.g., $X_{-k}$ denotes random variables $X_i$ for $i \in [p]$ except for $i = k$ and $x_{-k}$ denotes a matrix with all columns but the $k$-th. $I_n$ is the $n$-dimensional identity matrix. $P_{-k}$ denotes the orthogonal projection onto $x_{-k}$ and $P_{-k}^{\perp} = I_n - P_{-k}$ denotes the orthogonal projection onto its complement. $P_x$ is the orthogonal projection onto all $x$.

For some random vector $X$, we have the moment matrix $\epsilon^X := \mathbb{E}\left(XX^\top\right)$. This equals the covariance matrix for centered $X$. We assume this matrix to be invertible. Then, the principal submatrix $\epsilon_{-j,-j}^X := \mathbb{E}\left(X_{-j}X_{-j}^\top\right)$ is also invertible. We denote statistical independence by $\perp$. Besides, we use the superindex $k$ for all variables in the case of regression on substituted data. Besides, we use $\mathcal{O}_p(1)$ and $o_p(1)$ to denote the stochastic boundedness and stochastic convergence to 0, respectively.

**Theorem 1.** *For any $k \in [p]$, the cross-regression coefficient $\tilde{\beta}_{X \to Y}^{f,k}$ obtained from $[\tilde{X}_k, Y] \xrightarrow{OLS} f(Y)$, can be connected to the original regression coefficient $\beta_{X \to Y}^{f,OLS}$ obtained from $[X, Y] \xrightarrow{OLS} f(Y)$ as follows:*

$$(\tilde{W}^k)^T \tilde{\beta}_{X \to Y}^{f,k} = W^T \beta_{X \to Y}^{f,OLS}. \tag{6}$$

*Proof.* We can derive the original expression of $\beta_{X \to Y}^{f, OLS}$ as follows:

$$
\begin{aligned}
\beta_{X \to Y}^{f, OLS} &= E\left(XX^\top\right)^{-1} E\left\{Xf(Y)\right\} \\
&= \left(W^{-1}\right)^\top E\left(\Phi\Phi^\top\right)^{-1} W^{-1} W E\left\{\Phi f(Y)\right\} \\
&= \left(W^{-1}\right)^\top \operatorname{diag}\left\{\frac{1}{E\left(\Phi_1^2\right)}, \ldots, \frac{1}{E\left(\Phi_p^2\right)}\right\} E\left\{\Phi f(Y)\right\} \quad (7) \\
&= \left(W^{-1}\right)^\top \left[\frac{E\left\{\Phi_1 f(Y)\right\}}{E\left(\Phi_1^2\right)}, \ldots, \frac{E\left\{\Phi_p f(Y)\right\}}{E\left(\Phi_p^2\right)}\right]^\top,
\end{aligned}
$$

where the third equality follows from the independence of the $\Phi$. By analogy, we can derive the similar equality for $\tilde{\beta}_{X \to Y}^{f, k}$:

$$
\begin{aligned}
\tilde{\beta}_{X \to Y}^{f, k} &= E\left(\tilde{X}^k \left(\tilde{X}^k\right)^\top\right)^{-1} E\left\{\tilde{X}^k f(Y)\right\} \\
&= \left(\left(\tilde{W}^k\right)^{-1}\right)^\top E\left(\Phi\Phi^\top\right)^{-1} \left(\tilde{W}^k\right)^{-1} \tilde{W}^k E\left\{\Phi f(Y)\right\} \\
&= \left(\left(\tilde{W}^k\right)^{-1}\right)^\top \operatorname{diag}\left\{\frac{1}{E\left(\Phi_1^2\right)}, \ldots, \frac{1}{E\left(\Phi_p^2\right)}\right\} E\left\{\Phi f(Y)\right\} \quad (8) \\
&= \left(\left(\tilde{W}^k\right)^{-1}\right)^\top \left[\frac{E\left\{\Phi_1 f(Y)\right\}}{E\left(\Phi_1^2\right)}, \ldots, \frac{E\left\{\Phi_p f(Y)\right\}}{E\left(\Phi_p^2\right)}\right]^\top.
\end{aligned}
$$

Then combining the last equation in Eq.(7) and Eq.(8), the conclusion follows. $\square$

For the sake of clafity, we first prove a simplified version of Theorem 1 as follows:

**Theorem 2 (Simplified)** *Assume that the permutation $[p]$ is a causal order, i.e., $i \in An(j)$ once $i > j$ in $[p]$. Then for any $k \in [p]$ and $X_k \in An(Y)$, $X_k \in X^{RC}$ if and only if $\tilde{\beta}_Y^{f, k}[l] \neq \beta_Y^f[l]$ for each $X_l \in An(X_k)$.*

*Proof.* Throughout our proof, we use $\tilde{\beta}_{X \to Y}^{f, k}[l], \beta_{X \to Y}^{f, OLS}[l]$ and $\tilde{\beta}_{X \to Y}^{f, k}[l], \beta_{X \to Y}^{f, OLS}[l]$ interleavely by omitting the $X \to$ and putting $[l]$ into the subscript for convenience. We first build the connection between $W$ and $G$ (the same holds for $\tilde{W}$ and $\tilde{G}$):

$$
W_{ij} = Q_j \sum_{\substack{p \in Cp(j \to i) \\ p = X_j \to X_{j_1} \to \cdots \to X_{j_s} \to X_i}} G_{j_1, j} G_{j_2, j_1}, \ldots, G_{i, j_s}, \quad \forall X_j \in An(X_i) \quad (9)
$$

where $Cp(j \to i)$ denotes to the set of causal paths from $X_j$ to $X_i$. An immediate conclusion from Eq.(9) is that once a variable $X_l$ is soft-intervened, i.e., $\{Q_i\} \cup \{G_{kl}, X_l \in Pa(X_l)\}$ varies, then both $\{W_{kl}, X_l \in An(X_l)\}$ and $\{W_{ml}, X_m \in De(X_l), X_l \in An(X_l)\}$ might be vary. In other words, once $W_{kl}$ varies for $X_l \in An(X_l)$, either $X_l$ or some variable in $An(X_l)$ might be intervened.

Based on Theorem 1, we have:

$$
(\tilde{W}^k)^T \tilde{\beta}_{X \to Y}^{f, k} = W^T \beta_{X \to Y}^{f, OLS}, \quad (10)
$$

Recalling the fact that $\tilde{W}^k$ differs from $W$ only in the $k$-th row ($X_k = W_{k.}^T \Phi$ and $\tilde{X}_k = \tilde{W}_{k.}^T \Phi$), the matrix $(\tilde{W}^k)^T$ differs from $W^T$ only in the $k$-th column. Based on the fact that $[p]$ is a causal order, both $W$ and $\tilde{W}^k$ are lower-triangular matrices, i.e., both $W^T$ and $\left(\tilde{W}^k\right)^T$ are upper-triangular matrices.

To build the connection between the status of whether $X_l$ is intervened and the observed regression coefficients, i.e., $\tilde{\beta}_{X \to Y}^{f, k}$ and $\beta_{X \to Y}^{f, OLS}$, we divide the status of $X_l$ into four cases:

(1) $X_l$ is not intervened, and no causal paths exist between $X_l$ and intervened variables;

(2) $X_l$ is not intervened, and $X_l$ has no intervened ancestors;

(3) $X_l$ is not intervened, and $X_l$ has intervened ancestors;

(4) $X_l$ is intervened.

We first show that the first two cases are characterized in the following connection:

$\tilde{\beta}_{Y,l}^{f,X_k} = \beta_{X \to Y}^{f,OLS}[l]$ for each $l \in [p]$ if and only if $(\tilde{W}^k)^T = W^T$.

The proof is conducted in two folds:

- $\tilde{\beta}_{Y,l}^{f,X_k} = \beta_{X \to Y}^{f,OLS}[l]$ for each $l \in [p]$. Then it is easy to show that $(\tilde{W}^k)^T = W^T$. To be specific, considering the case when $l = p$, i.e., only $(\tilde{W}^k)_{pp}^T \neq 0$ in the $p$-th row of $(\tilde{W}^k)^T$, then we immediately conclude that $(\tilde{W}^k)_{pp}^T = (W^k)_{pp}^T$, i.e., $(\tilde{W}^k)_{p.}^T = (W^k)_{p.}^T$ By analogy, based on $(\tilde{W}^k)_{p.}^T = (W^k)_{p.}^T$, we can conclude that $(\tilde{W}^k)_{p-1,.}^T = (W^k)_{p-1,.}^T$. The rest can be derived in the same manner.

- $(\tilde{W}^k)^T = W^T$. Then it is easy to show that $\tilde{\beta}_{Y,l}^{f,X_k} = \beta_{X \to Y}^{f,OLS}[l]$ for each $l \in [p]$. To be specific, considering the case when $l = p$, i.e., only $(\tilde{W}^k)_{pp}^T \neq 0$ in the $p$-th row of $(\tilde{W}^k)^T$, then we immediately conclude that $\tilde{\beta}_{Y,p}^{f,X_k} = \beta_{Y,p}^f$. By analogy, based on $\tilde{\beta}_{Y,p}^{f,X_k} = \beta_{Y,p}^f$, we can conclude that $\tilde{\beta}_{Y,p-1}^{f,X_k} = \beta_{Y,p-1}^f$. The rest can be derived in the same manner.

Based on the above conclusion, when $\tilde{\beta}_{Y,l}^{f,X_k} = \beta_{X \to Y}^{f,OLS}[l]$ for each $l \in [p]$, we can conclude that $X_l$ is not intervened, i.e., $X_l \notin X^{RC}$, based on the regularity condition in Eq.(1). For the case (1), according to the expression in Eq.(9), $W_{kl}$ will not vary for each $X_l \in An(X_l)$, hence $(\tilde{W}^k)^T = W^T$. By analogy, in case (2), $W_{kl}$ will not change according to Eq.(9).

Subsequently, when $\tilde{\beta}_{Y,l}^{f,X_k} = \beta_{X \to Y}^{f,OLS}[l]$ varies for some $l \in [p]$, we prove our final goal, i.e., distinguish case (4) from case (3) using the following criteria:

$\tilde{\beta}_{Y,l}^{f,X_k} \neq \beta_{X \to Y}^{f,OLS}[l]$ for each $X_l \in An(X_l)$ if and only if $X_l$ is intervened

- We first show the sufficiency by contradiction, i.e., $X_l$ is not intervened. We prove this by checking variables from $p$ to 1 in the causal order. Since $W_{q.}^T = (\tilde{W}^k)_{q.}^T$ for each $q \notin An(X_l)$ ($q > k$ in $[p]$ in upper-triangular matrices), we have $\tilde{\beta}_{Y,q}^{f,X_k} = \beta_{Y,q}^f$ based on existing conditions $\left(W_{q.}^T\right)^T \beta_{Y,q}^f = \left((\tilde{W}^k)_{q.}^T\right)^T \tilde{\beta}_{X \to Y}^{f,k}$. As $(\tilde{W}^k)_{kk}^T = \tilde{Q}_l$ and $W_{kk}^T = Q_l$ ($\tilde{Q}_l = Q_l$), we have $\tilde{\beta}_{Y,k}^{f,X_k} = \beta_{Y,k}^f$. Now we check each variable $X_l$ with $l <$ in $[p]$ and $X_l \notin X_k$. Recalling that both $(\tilde{W}^k)^T$ and $W^T$ are upper-triangular matrices and $[p]$ is a causal order, we observe that both $W_{lk}^T = 0$ and $(\tilde{W}^k)_{lk}^T = 0$ (According to Eq.(9)). Hence, we can conclude that $\tilde{\beta}_{Y,l}^{f,X_k} \neq \beta_{X \to Y}^{f,OLS}[l]$. Combining with the previous conclusion that $\tilde{\beta}_{Y,k}^{f,X_k} = \beta_{Y,k}^f$, when checking the first parental variable $X_l$ with $l < k$ in $[p]$, we have $\tilde{\beta}_{Y,l}^{f,X_k} = \beta_{X \to Y}^{f,OLS}[l]$. Hence, we further conclude that $\tilde{\beta}_{Y,k}^{f,X_k} = \beta_{Y,k}^f$ for any $X_l \in Pa(X_l)$. Then the sufficiency follows.

- The necessity is immediate when we check the definition of soft-intervention and the expression in Eq.(9).

$\square$

**Theorem 2.** *For any $k \in [p]$ and $X_k \in An(Y)$, $X_k \in X^{RC}$ if and only if $\tilde{\beta}_{Y}^{f,k}[l] \neq \beta_Y^f[l]$ for each $X_l \in An(X_k)$.*

*Proof.* We then finalize our proof on the main theorem in our paper. As already shown in the above Theorem with $[p]$ a causal order, our claim holds. Recalling the equation as follows:

$$(\tilde{W}^k)^T \tilde{\beta}_{X \to Y}^{f,k} = W^T \beta_{X \to Y}^{f,OLS}, \tag{11}$$

with the fact that $\tilde{W}^k$ and $W^T$ can be turned into upper-triangular matrices by row-wise permutations when $[p]$ is not a causal order. Hence, by simultaneously multiply the same permutation matrix $P^2$, we obtain the following equation:

$$(\tilde{W}^k)_{\pi(p)}^T \tilde{\beta}_{X \to Y}^{f,k} = P(\tilde{W}^k)^T \tilde{\beta}_{X \to Y}^{f,k} = PW^T \beta_{X \to Y}^{f,OLS} = W_{\pi(p)}^T \beta_{X \to Y}^{f,OLS}, \tag{12}$$

where $(\tilde{W}^k)_{\pi(p)}^T$ and $W_{\pi(p)}^T$ denotes the permuted upper-triangular matrices. Then the case is boiled down to the case of the previous Theorem, and the claim follows. $\square$

We then introduce some basic definitions with notations from previous advances in partial regression theory Schultheiss et al. (2024) as follows:

$$\begin{aligned}
Z_l &:= X_l - X_{-l}^\top \gamma_l, \\
\gamma_l &:= \underset{b \in \mathbb{R}^{p-1}}{\operatorname{argmin}} \mathbb{E}\left\{\left(X_l - X_{-l}^\top b\right)^2\right\} = \left(\epsilon_{-l,-l}^X\right)^{-1} \mathbb{E}\left(X_{-l} X_l\right), \\
W_l &:= f(Y) - X_{-l}^\top \zeta_l, \\
\zeta_l &:= \underset{b \in \mathbb{R}^{p-1}}{\operatorname{argmin}} \mathbb{E}\left\{\left(f(Y) - X_{-l}^\top b\right)^2\right\} = \left(\epsilon_{-l,-l}^X\right)^{-1} \mathbb{E}\left\{X_{-l} f(Y)\right\}
\end{aligned} \tag{13}$$

Using these definitions, previous advances have shown that $\beta_{X \to Y}^{f,OLS}[l] = \mathbb{E}(Z_l W_l)/\mathbb{E}[Z_l^2] = \mathbb{E}[Z_l f(Y)]/\mathbb{E}[Z_l^2]$ from partial regression Schultheiss et al. (2024); Schultheiss & Bühlmann (2023).

**Remark 3.** *We note that the above defined $Z_l$, $\gamma_l$, $W_l$ and $\zeta_l$ with the equation that $\beta_{X \to Y}^{f,OLS}[l] = \mathbb{E}[Z_l W_l]/\mathbb{E}[Z_l^2] = \mathbb{E}[Z_l f(Y)]/\mathbb{E}[Z_l^2]$ also holds for substituted cases, i.e., for $Z_l^k$, $\gamma_l^k$, $W_l^k$ and $\zeta_l^k$ and $\tilde{\beta}_{X \to Y}^{f,k}[l] = \mathbb{E}[Z_l^k W_l^k]/\mathbb{E}[(Z_l^k)^2] = \mathbb{E}[Z_l^k f(Y)]/\mathbb{E}[(Z_l^k)^2]$.*

We first define some empirical statistics as follows Schultheiss & Bühlmann (2023):

$$\hat{z}_l = P_{-l}^\perp x_l \quad \text{and} \quad \hat{w}_l = P_{-l}^\perp f(y) \quad \text{such that} \quad \hat{\beta}_{Y,l}^f = \frac{\hat{z}_l^\top \hat{w}_l}{\hat{z}_l^\top \hat{z}_l}. \tag{14}$$

Then the following lemma holds in previous advances Schultheiss & Bühlmann (2023):

**Lemma 2.**

$$\begin{aligned}
\frac{1}{n}\hat{z}_l^\top \hat{w}_l &= \mathbb{E}(Z_l W_l) + o_p(1) = \frac{1}{n}z_l^\top w_l + o_p\left(\frac{1}{\sqrt{n}}\right), \\
\frac{1}{n}\hat{z}_l^\top \hat{z}_l &= \mathbb{E}(Z_l^2) + o_p(1) = \frac{1}{n}z_l^\top z_l + o_p\left(\frac{1}{n}\right). \\
\left\| n\left(x_{-l}^\top x_{-l}\right)^{-1}\right\| &\xrightarrow{\mathbb{P}} \left\|\left(\epsilon_{-l,-l}^X\right)^{-1}\right\| = \mathcal{O}(1), \\
\left\| n\left(x^\top x\right)^{-1}\right\| &\xrightarrow{\mathbb{P}} \left\|\left(\epsilon^X\right)^{-1}\right\| = \mathcal{O}(1),
\end{aligned} \tag{15}$$

**Lemma 3.**

$$W_l = Z_l \beta_{X \to Y}^{f,OLS}[l] + \mathcal{E}. \tag{16}$$

**Remark 4.** *We note that the above definitions, Lemma 2, and Lemma 3 also holds for the substituted variables, i.e., $\hat{z}_l^k$, $\hat{w}_l^k$, $Z_l^k$ and $W_l^k$.*

We then present several lemmas to sharpen some convergence results based on the above existing lemmas.

---

[2]The definition of soft intervention and the parental set do not increase guarantee this operation.

**Lemma 4** (Variant of convergence of $\frac{1}{n}\hat{z}_l^\top \hat{w}_l$.). *By assuming that $W_l$, $Z_l$ and $X_l$ are fourth-moment bounded, i.e., $\mathbb{E}[Z_l^4] < \infty$, $\mathbb{E}[W_l^4] < \infty$ and $\mathbb{E}[X_l^4] < \infty$, we have:*

$$\frac{1}{n}\hat{z}_l^\top \hat{w}_l = \frac{1}{n}z_l^\top w_l + \mathcal{O}_p(\frac{1}{n}). \tag{17}$$

*Proof.* We first expand the expression of the target term as follows:

$$
\begin{aligned}
\left| z_l^\top P_{-l} w_l \right| &= \left| z_l^\top x_{-l} \left( x_{-l}^\top x_{-l} \right)^{-1} x_{-l}^\top w_l \right| \\
&\leq \left\| z_l^\top x_{-l} \right\|_2 \left\| \left( x_{-l}^\top x_{-l} \right)^{-1} \right\|_2 \left\| x_{-l}^\top w_l \right\|_2 \\
&\leq \left\| z_l^\top x_{-l} \right\|_1 \left\| \left( x_{-l}^\top x_{-l} \right)^{-1} \right\|_2 \left\| x_{-l}^\top w_l \right\|_1 \\
&= \sum_{k \neq l} \left| z_l^\top x_k \right| \left\| \left( x_{-l}^\top x_{-l} \right)^{-1} \right\|_2 \sum_{k \neq l} \left| x_k^\top w_l \right|,
\end{aligned}
\tag{18}
$$

Letting $T_{kjl} = z_{lj} x_{kj}$, we further expand the term $\sum_{k \neq l} \left| z_l^\top x_k \right|$ as follows:

$$\sum_{k \neq l} \left| z_l^\top x_k \right| = \sum_{k \neq l} \sum_j T_{kjl}. \tag{19}$$

By treating the term $\sum_j T_{kjl}$ as the sum of i.i.d. copies, the Chebyshev's Inequality entails that:

$$P(|\sum_j T_{kjl} - nE[X_l Z_l]| \geq c) \leq \frac{n\mathrm{Var}(T_{kjl})}{c^2}, \tag{20}$$

where we note that $\mathbb{E}[X_k Z_l] = 0$ and $\mathbb{E}[W_l X_k] = 0$ due to the fact that $X_{-l}^\top \gamma_l$ and $X_{-l}^\top \zeta_l$ are $\mathcal{L}^2$ projections of $X_l$ and $f(Y)$ on the subspace spanned by $X_{-l}$, respectively. Hence, the above inequality can be simplified as follows:

$$P(|\sum_j T_{kjl}| \leq c) \geq \frac{n\mathbb{E}[Z_l^2 X_k^2]}{c^2} \Rightarrow \sum_{k \neq l} |\sum_j T_{kjl}| = \mathcal{O}(\sqrt{n}). \tag{21}$$

In analog, we can derive that $\sum_{k \neq l} \left| x_l^\top w_l \right| = \mathcal{O}(\sqrt{n})$. By combining with the fact that $\left\| n \left( x_{-l}^\top x_{-l} \right)^{-1} \right\| \overset{\mathbb{P}}{\to} \left\| \left( \epsilon_{-l,-l}^X \right)^{-1} \right\| = \mathcal{O}(1)$ in Lemma 2, we have:

$$\sum_{k \neq l} \left| z_l^\top x_k \right| \left\| \left( x_{-l}^\top x_{-l} \right)^{-1} \right\|_2 \sum_{k \neq l} \left| x_k^\top w_l \right| = \mathcal{O}(\sqrt{n})\mathcal{O}(\frac{1}{n})\mathcal{O}(\sqrt{n}) = \mathcal{O}(1). \tag{22}$$

Then our claim holds by the following derivation:

$$
\begin{aligned}
\frac{1}{n}\hat{z}_l^\top \hat{w}_l &= \frac{1}{n}z_l^\top P_{-l}^\perp w_l \\
&= \frac{1}{n}\left( z_l^\top w_l - z_l^\top P_{-l} w_l \right) \\
&= \frac{1}{n}z_l^\top w_l + \mathcal{O}_p\left( \frac{1}{n} \right) \\
&= \frac{1}{n}z_l^\top w_l + o_p\left( \frac{1}{n} \right).
\end{aligned}
\tag{23}
$$

$\square$

**Lemma 5** (Variant of convergence of $\frac{1}{n}\hat{z}_l^\top \hat{z}_l$.). *By assuming that $Z_l, W_l$ is square-integrable, i.e., $\mathbb{E}[Z_l^2] < \infty$ and $\mathbb{E}[W_l^2] < \infty$, we have:*

$$
\begin{aligned}
\frac{1}{n}\hat{z}_l^\top \hat{z}_l &= \mathbb{E}\left( Z_l^2 \right) + o_p(\frac{1}{\sqrt{n}}), \\
\frac{1}{n}\hat{z}_l^\top \hat{w}_l &= \mathbb{E}\left( Z_l W_l \right) + o_p(\frac{1}{\sqrt{n}}),
\end{aligned}
\tag{24}
$$

*Proof.* We first prove a pre-conclusion in the case that the variables $Z_l^2$ is centered, i.e., $\mathbb{E}[Z_l^2] = 0$. By the Marcinkiewicz-Zygmund Theorem (cf. Y. S. Chow & H. Teicher, Probability Theory, 3Ed, Springer Verlag, 1997, p.125, Theorem 2.5.2), we have :

$$\frac{\sum_{l=1}^n z_l^2}{\sqrt{n}} \xrightarrow{\text{a.s.}} 0. \tag{25}$$

Hence, we have:

$$\frac{\sum_{l=1}^n z_l^2}{n} = \frac{\sum_{l=1}^n z_l^2}{\sqrt{n}} \frac{1}{\sqrt{n}} = \mathcal{O}(\frac{1}{\sqrt{n}}) = \mathcal{O}_p(\frac{1}{\sqrt{n}}).$$

When $Z_l^2$ is not centered, we immediately obtain that:

$$\frac{\sum_{l=1}^n Z_l^2}{n} = \mathbb{E}[Z_l^2] + \mathcal{O}_p(\frac{1}{\sqrt{n}}).$$

Hence, combining with the fact that $\frac{1}{n}\hat{z}_l^\top \hat{z}_l = \frac{1}{n}z_l^\top z_l + \mathcal{O}_p(\frac{1}{n})$ in Lemma 2, the claim holds (Cauchy–Schwarz inequality entails that $\mathbb{E}[Z_l^4] < \infty$ by condition).

The second conclusion follows by an analog. $\qquad\square$

**Lemma 6** (Variant of convergence of $\frac{1}{n}(\hat{z}_l^k)^\top \hat{z}_l^k$ and $\frac{1}{n}(\hat{z}_l^k)^\top \hat{w}_l^k$.). *By assuming that $Z_l^k, W_l^k$ is square-integrable, i.e., $\mathbb{E}[(Z_l^k)^2] < \infty$ and $\mathbb{E}[(W_l^k)^2] < \infty$, we have:*

$$\frac{1}{n}(\hat{z}_l^k)^\top \hat{z}_l^k = \mathbb{E}[(Z_l^k)^2] + \mathcal{O}_p(\frac{1}{\sqrt{n}})$$
$$\frac{1}{n}\hat{z}_l^\top \hat{w}_l = \mathbb{E}(Z_l W_l) + \mathcal{O}_p(\frac{1}{\sqrt{n}}) \tag{26}$$
$$\frac{1}{n}(\hat{z}_l^k)^\top \hat{w}_l^k = \frac{1}{n}(z_l^k)^\top w_l^k + \mathcal{O}_p(\frac{1}{n}).$$

*Proof.* The outline of our proof keeps the same as the above two lemmas. $\qquad\square$

**Theorem 3.** *Assume that $E\left\{f(Y)^2\right\} < \infty, E(X_k^4) < \infty$ for all $k$, and $\tilde{\beta}_{X \to Y}^{f,k}[l]$ and $\beta_{X \to Y}^{f,OLS}[l]$ exists. Then for variables $X_k$ in $An(X_l)$ with $\tilde{\beta}_{X \to Y}^{f,k}[l] = \beta_{X \to Y}^{f,OLS}[l]$, the following convergence in distribution holds:*

$$\sqrt{n}\left(\widehat{\tilde{\beta}_{X \to Y}^{f,k}} - \widehat{\beta_{X \to Y}^{f,OLS}}[l]\right) \xrightarrow{\mathbb{D}} \mathcal{N}\left(0, \mathbb{E}\left[\mathcal{E}^2\right] + \mathbb{E}\left[(\mathcal{E}^k)^2\right]\right) \tag{27}$$

*Proof.* We first expand the target term as follows:

$$\widehat{\tilde{\beta}_{X \to Y}^{f,k}}[l] - \widehat{\beta_{X \to Y}^{f,OLS}}[l] = \frac{\frac{1}{n}(\hat{z}_l^k)^\top \hat{w}_l^k}{\frac{1}{n}(\hat{z}_l^k)^\top \hat{z}_l^k} - \frac{\frac{1}{n}(\hat{z}_l)^\top \hat{w}_l}{\frac{1}{n}(\hat{z}_l)^\top \hat{z}_l}, \tag{28}$$

where we then expand each term in equation 28 as follows:

$$\sqrt{n}\widehat{\beta_{X\to Y}^{f,OLS}}[l] = \frac{\sqrt{n}\frac{1}{n}\hat{z}_l^\top \hat{w}_l}{\frac{1}{n}\hat{z}_l^\top \hat{z}_l}$$

$$= \frac{\sqrt{n}\frac{1}{n}z_l^\top w_l + \mathcal{O}_p(1/\sqrt{n})}{\frac{1}{n}z_l^\top z_l + \mathcal{O}_p(1/n)}$$

$$= \frac{\sqrt{n}\frac{1}{n}z_l^\top w_l}{\frac{1}{n}z_l^\top z_l + o_p(1)} + \frac{\mathcal{O}_p(1/\sqrt{n})}{\mathbb{E}\left[Z_l^2\right] + o_p(1)}$$

$$= \frac{\sqrt{n}\frac{1}{n}z_l^\top w_l}{\frac{1}{n}z_l^\top z_l} + \frac{\mathcal{O}_p(1/\sqrt{n})}{\mathcal{O}_p(1)}$$

$$= \sqrt{n}\beta_{X\to Y}^{f,OLS}[l] + \frac{\sqrt{n}\frac{1}{n}z_l^\top \epsilon}{\frac{1}{n}z_l^\top z_l} + \mathcal{O}_p(1/\sqrt{n}) \qquad (29)$$

$$= \sqrt{n}\beta_{X\to Y}^{f,OLS}[l] + \frac{\sqrt{n}\frac{1}{n}z_l^\top \epsilon}{\mathbb{E}\left[Z_l^2\right] + o_p(1/\sqrt{n})} + \mathcal{O}_p(1/\sqrt{n})$$

$$= \sqrt{n}\beta_{X\to Y}^{f,OLS}[l] + \frac{\sqrt{n}\frac{1}{n}z_l^\top \epsilon}{\mathbb{E}\left[Z_l^2\right] + \mathcal{O}_p(1/\sqrt{n})} + \mathcal{O}_p(1/\sqrt{n})$$

$$= \sqrt{n}\beta_{X\to Y}^{f,OLS}[l] + \frac{\sqrt{n}\frac{1}{n}z_l^\top \epsilon}{\mathbb{E}\left[Z_l^2\right]} + \mathcal{O}_p(1/\sqrt{n})$$

$$= \sqrt{n}\beta_{X\to Y}^{f,OLS}[l] + \frac{\sqrt{n}\frac{1}{n}z_l^\top \epsilon}{\mathbb{E}\left[Z_l^2\right]} + o_p(1),$$

where we invoke Lemma 4 in the second equality, and invoke Lemma 3 in the fourth equality.

By an analog, we invoke the Lemma 6 and derive the following equation:

$$\sqrt{n}\widehat{\tilde{\beta}_{X\to Y}^{f,k}}[l] = \sqrt{n}\tilde{\beta}_{X\to Y}^{f,k}[l] + \frac{\sqrt{n}\frac{1}{n}(z_l^k)^\top \epsilon^k}{\mathbb{E}\left[(Z_l^k)^2\right]} + o_p(1), \qquad (30)$$

By combining the above two expressions, we obtain that:

$$\sqrt{n}\left(\widehat{\beta_{X\to Y}^{f,OLS}}[l] - \widehat{\tilde{\beta}_{X\to Y}^{f,k}}[l]\right) = \frac{1}{\sqrt{n}}\left(\frac{z_l^\top \epsilon}{\mathbb{E}\left[Z_l^2\right]} - \frac{(z_l^k)^\top \epsilon^k}{\mathbb{E}\left[(Z_l^k)^2\right]}\right) + o_p(1). \qquad (31)$$

Finally, by the CLT theorem, we derive the convergence in distribution as follows:

$$\sqrt{n}\left(\widehat{\beta_{X\to Y}^{f,OLS}}[l] - \widehat{\tilde{\beta}_{X\to Y}^{f,k}}[l]\right) \xrightarrow{\mathbb{D}} \mathcal{N}\left(\mathbb{E}\left[\left(\frac{Z_l\mathcal{E}}{\mathbb{E}[Z_l^2]} - \frac{Z_l^k\mathcal{E}^k}{\mathbb{E}\left[(Z_l^k)^2\right]}\right)\right], \operatorname{var}\left(\frac{Z_l\mathcal{E}}{\mathbb{E}\left[Z_l^2\right]} - \frac{Z_l^k\mathcal{E}^k}{\mathbb{E}\left[(Z_l^k)^2\right]}\right)\right), \qquad (32)$$

As $\beta_{X\to Y}^{f,OLS}[l] = \tilde{\beta}_{X\to Y}^{f,k}[l]$ entails that $\frac{\mathbb{E}[Z_l\mathcal{E}]}{\mathbb{E}[Z_l^2]} - \frac{\mathbb{E}[Z_l^k\mathcal{E}^k]}{\mathbb{E}\left[(Z_l^k)^2\right]} = 0$, the limit distribution is centered. Furthermore, we expand the variance term as follows:

$$\operatorname{var}\left(\frac{Z_l\mathcal{E}}{\mathbb{E}\left[Z_l^2\right]} - \frac{Z_l^k\mathcal{E}^k}{\mathbb{E}\left[(Z_l^k)^2\right]}\right)$$

$$= \mathbb{E}\left[\left(\frac{Z_l\mathcal{E}}{\mathbb{E}\left[Z_l^2\right]}\right)^2\right] + \mathbb{E}\left[\left(\frac{Z_l^k\mathcal{E}^k}{\mathbb{E}\left[(Z_l^k)^2\right]}\right)^2\right] - 2\mathbb{E}\left[\left(\frac{Z_l\mathcal{E}}{\mathbb{E}\left[Z_l^2\right]}\right)\right]\mathbb{E}\left[\left(\frac{Z_l^k\mathcal{E}^k}{\mathbb{E}\left[(Z_l^k)^2\right]}\right)\right] \qquad (33)$$

$$= \mathbb{E}\left[\left(\frac{\mathbb{E}\left[Z_l^2\mathcal{E}^2\right]}{\mathbb{E}\left[Z_l^2\right]}\right)^2\right] + \mathbb{E}\left[\left(\frac{\mathbb{E}\left[(Z_l^k)^2(\mathcal{E}^k)^2\right]}{\mathbb{E}\left[(Z_l^k)^2\right]}\right)^2\right]$$

$$= \mathbb{E}\left[\mathcal{E}^2\right] + \mathbb{E}\left[(\mathcal{E}^k)^2\right],$$

where the first equality and third equality are due to the fact that $Z_l \perp\!\!\!\perp \mathcal{E}$, $Z_l^k \perp\!\!\!\perp \mathcal{E}$, and $\mathcal{E} = \mathcal{E}^{\parallel} = 0$ (the zero-expected residual can be easily achieved by adding an intercept when performing regression.) $\qquad\square$

**Theorem 4.**[*Variance Estimation*] *Assume that , the following convergence in probability holds:*

$$(\hat{\epsilon})^2 + \left(\hat{\epsilon^k}\right)^2 \xrightarrow{\mathbb{P}} \mathbb{E}\left[\mathcal{E}^2\right] + \mathbb{E}\left[(\mathcal{E}^k)^2\right], \tag{34}$$

*where* $(\hat{\epsilon})^2 := \frac{\|f(y)-x\widehat{\beta_{X\to Y}^{f,OLS}}\|_2^2}{n-p}$ *and* $\left(\hat{\epsilon^k}\right)^2 := \frac{\|f(y)-x^k\widehat{\bar{\beta}_{X\to Y}^{f,k}}\|_2^2}{n-p}$.

*Proof.* We prove the above theorem by showing the convergence of $(\hat{\epsilon})^2$:

$$(\hat{\epsilon})^2 := \frac{\|f(y) - x\widehat{\beta_{X\to Y}^{f,OLS}}\|_2^2}{n-p} = \frac{\hat{\epsilon}^\top \hat{\epsilon}}{n-p} = \frac{\epsilon^\top P_x^\perp \epsilon}{n-p} = \frac{\epsilon^\top \epsilon - \epsilon^\top P_x \epsilon}{n-p}, \tag{35}$$

where we can derive the following equation with some basic properties of orthogonal projection matrix:

$$\begin{aligned}
\hat{\epsilon} - P_x^\perp \epsilon &= f(y) - x\widehat{\beta_Y^f} - P_x^\perp(f(y) - x\beta_Y^f) \\
&= P_x f(y) - x\widehat{\beta_Y^f} + P_x^\perp x\beta_Y^f \\
&= P_x f(y) - P_x x\widehat{\beta_Y^f} \\
&= 0.
\end{aligned} \tag{36}$$

Recalling equation 35, we further have that:

$$\begin{aligned}
\frac{\epsilon^\top P_x \epsilon}{n-p} &= \frac{\epsilon^\top x(x^\top x)^{-1}x^\top \epsilon}{n-p} \\
&\leq \frac{\|\epsilon^\top x\|_2 \|(x^\top x)^{-1}\|_2 \|x^\top \epsilon\|_2}{n-p} \\
&\leq \frac{\|\epsilon^\top x\|_1 \|(x^\top x)^{-1}\|_2 \|x^\top \epsilon\|_1}{n-p}, \\
&\leq \frac{\sum_l |\epsilon^\top x_l| \|(x^\top x)^{-1}\|_2 \sum_l |\epsilon^\top x_l|}{n-p} \\
&= \frac{\mathcal{O}_p(n)\mathcal{O}_p(\frac{1}{n})\mathcal{O}_p(n)}{n-p} = \mathcal{O}_p(1).
\end{aligned} \tag{37}$$

Hence, we obtain that:

$$(\hat{\epsilon})^2 = \frac{\epsilon^\top \epsilon}{n-p} + \mathcal{O}_p(1) = \mathbb{E}[\epsilon^2] + \mathcal{O}_p(1). \tag{38}$$

By analog, one can derive that:

$$\left(\hat{\epsilon^k}\right)^2 = \frac{(\epsilon^k)^\top \epsilon^k}{n-p} + \mathcal{O}_p(1) = \mathbb{E}[(\epsilon^k)^2] + \mathcal{O}_p(1). \tag{39}$$

, and our claim follows.

$\qquad\square$

# F    COMPLEXITY ANALYSIS

In addition to the main paper, we also analyze the complexity of the ASD pre-step in below:

**Proposition 1.** *The upper bound of the computational complexity of ASD sub-algorithm is* $\mathcal{O}(np^4)$.

| METHOD | CAUSAL GRAPH | SCM | INTERVENTION | CONFOUNDER | RETURN | METRIC | Kind |
|---|---|---|---|---|---|---|---|
| Travesal | Given | No | Hard | No | Top-K Recommend | Recall@K | Discovery |
| DeepITE | Unknown | No | Soft&Hard | No | Top-K Recommend | Recall@K | Discovery-based INI |
| UT-IGSP | Unknown | No | Soft | No | Fixed Identification | Precision | Discovery-based INI |
| CF-Attr | Given | Require | Structure-Preserving | No | Top-K Recommend | Recall@K | Counterfactual-based RCA |
| CIRCA | Given | No | Hard | No | Top-K Recommend | Recall@K | Discovery-based RCA |
| RCD | Unknown | No | Hard | No | Top-K Recommend | Recall@K | Discovery-based INI |
| TOCA | Given | No | Structure-Preserving | No | Top-K Recommend | Recall@K | Counterfactual-based RCA |
| IDI | Given | Require | Soft | No | Top-K Recommend | Recall@ | Statistics-based INI |
| LinearEST | Unknown | No | Soft | No | Fixed Identification | Precision | Statistics-based INI |
| iSCAN | Unknown | No | Soft | No | Fixed Identification | Precision | Statistics-based INI |
| CRRC (Ours) | Unknown | No | Soft | No | Fixed Identification | Precision | Statistics-based RCA |

Table 5: Comparison and introduction for each baseline.

*Proof.* First, we show that the upper bound of the number of performing OLS regression is $\mathcal{O}(p^2)$. To see this, it is obvious that the maximum number of $|An(Y)|$ is $p$. Consequently, without loss of generality, we assume that the index $1, 2, \ldots, p$ represents variables with $|An(X_i)|$ in descending order. It is obvious to see that $|An(X_1)| \leq (p-1)$. Hence, we have $|An(X_2)| \leq (p-2)$ (otherwise a cycle will exist). By analogy, we have:

$$\sum_{i=1}^{p} |An(X_i)| \leq \sum_{i=1}^{p-1} = \frac{p(p-1)}{2},$$

and the conclusion follows by considering that the complexity of OLS regression is $\mathcal{O}(np^2)$. □

# G  ADDITIONAL EXPERIMENTAL DETAILS

## G.1  EXPERIMENT SETUP

**Baseline Implementation.** To facilitate a fair comparison, the prior causal graph for each baseline are searched using the LiNGAM (Shimizu, 2014), eschewing the need for graph construction for Traversal, Eplison, RW, CIRCA, CF-Attr, TOCA and IDI. Meanwhile, as counterfactual-based RCA methods (except for TOCA) requires the whole SCM models, we also note that the LiNGAM method can recover the linear SCM (Shimizu, 2014) such that CF-Attr and IDI can obtain SCM models. Regarding implementations, we exactly follow tuning protocols of each method as follows:

- CIRCA with implementation in `https://github.com/NetManAIOps/CIRCA`;

- Traversel, RW, eplison, IDI,TOCA, CF-Attr, with implementation in the anonymous link of `https://openreview.net/forum?id=l11DZY5Nxu` in `https://anonymous.4open.science/r/petshop-BB8A/rca_task.py`;

- DeepITE with implementation in `https://github.com/antgroup/deepite`;

- Linear EST with implementation in `https://github.com/bvarici/intervention-estimation`;

- iSCAN with implementation in `https://github.com/kevinsbello/iscan`;

- RCD with implementation in `https://github.com/azamikram/rcd`;

- UT-IGSP with implementation in `https://github.com/csquires/utigsp`.

**Computational Recourses.** All the training & inference runs on 8 NVIDIA A100 GPUs with 40GB of VRAM.

**Metrics.** For methods identifying a fixed set (CRRC (ours), LinearEST, UT-IGSP, iSCAN), we choose the precision to measure the proportion of true RCs that are successfully captured within the grounding set of all RCs. For methods recommending top-K variables (the rest methods), we choose the Top-K Recall (Recall@K) to measure the proportion of true RCs that are successfully captured within the top-k ranked candidates proposed by each method.

## G.2 Description of Datasets

**Synthetic Dataset.** Following (Varici et al., 2021), we generate 100 realizations of Erdos-Rényi (ER) graph with expected neighborhood'size as 2, and Scale-free (SF) graph with expected outgoing edges'size as 100. We denote the generated random graph as $\mathcal{G}$. We then sample the entries of $\mathcal{G}$, i.e., the adjacency matrix's entries, from a uniform distribution on $[-2, 0) \cup (0, 2]$. Meanwhile, we sample each exogenous noise variable with equal probability from the Gaussian $\mathcal{N}(0, 1)$, exponential $\text{Exp}(1)$, uniform $U(0, 1)$, and gumbel distributions $\text{Gum}(1)$. As $X$ is already generated, we then generate the outcome $Y$ with exact the same generation protocol as in $X$.

We then generate the interventional/abnormal data. For intervened variable $X_k$, we shift both the exogenous noise $\Phi_k$ by increasing 1 on its variance and the edges weight from $Pa(X_k)$ from $[-2, 0) \cup (0, 2]$ to $U([-1, 1) \cup (0, 2])$. Finally, we generate the intervened ancestors with number $K$ from $Y$ using the following procedure:

(1) We rank all ancestors of $Y$ based on their ancestor effects on $Y$;

(2) We select the top $int(K/4)$ ancestors and the last $int(K/4)$ ancestors to comprise the grounding $X^{rc}$. To emphasize our problem setup, we let nearly half of the intervened variables, i.e., $int(K/2)$ variables not in $An(Y)$, also be intervened. Such a generation corresponds to the realistic cases that some intervened variables will cause other outcome variable we focus on.

We set the overall sample size to 2000 throughout our synthetic experiments. We set $K = 8$ throughout our experiments, except for our robustness testing on the RC number.

**Semi-synthetic Supply-chain Dataset.** We extend the semi-synthetic supply chain dataset (Budhathoki et al., 2022; Nguyen et al., 2024), i.e., a case study simulating the real-world supply-chain system. In supply chain management, the availability of product units in inventory, ready for shipment, plays a pivotal role in meeting customer demand promptly `https://www.pywhy.org/dowhy/v0.9.1/example_notebooks/gcm_supply_chain_dist_change.html`. Consequently, retailers consistently procure products in anticipation of future customer requirements. Consider a scenario where a retailer submits weekly purchase orders (POs) to vendors, factoring in projected future demand and existing capacity constraints. Vendors subsequently evaluate these orders and confirm whether they can fulfill them partially or entirely. Upon mutual agreement between the vendors and the retailer, the products are dispatched. However, it is important to note that the entirety of the confirmed POs may not be delivered simultaneously. As shown in Fig. 6, we extend 5 demand variables and 5 constraints to simulate diverse demand forecasting and constraint capacity in realistic supply-chain scenes. Following (Budhathoki et al., 2022), we simulate exogenous variables for each demand and constraint variable as $\text{Gamma}(1)$, where the scale parameter is set to 1. Meanwhile, each edge weights are sampled from $U([-2, -1] \cup [1, 2])$. To simulate the interventional data, we shift weight from ancestor to each intervened variables by increasing from $U([-2, -1] \cup [1, 2])$ to from $U([-1, 0) \cup [2, 3])$. We realize 100 cases by randomly selecting the root cause set. To be specific, we select $X^{RC}$ with number K as the combination of $int(K/2)$ of the ["D1","D3","C1","C3"] and $int(K/2)$ from ["D2","D4", "D5", "C2","C4", "C5"]. We set $K = 4$ throughout our experiments.

**Real-world Protein outcome variable Dataset.** The widely recognized protein signaling dataset, originally introduced by (Sachs et al., 2005), explores the intricate interactions within T-cell signaling networks. This dataset encompasses 11 nodes and 16 edges, comprising 1,755 observational samples and 4,091 interventional samples obtained across five distinct experimental conditions involving various drug interventions to modulate signaling proteins. We harness an accepted ground truth network structure in (Ness et al., 2018), and the preprocessing steps outlined in (Tao et al., 2024), to benchmark CRRC's performance with other models. We note that, as our proposed CRRC method requires the outcome variable, we manually generate the outcome variable by simply adding 11 variable together. **In such construction of the outcome variable, the RCA problem reduces to the trivial INI problem, where all variables are ancestors of the outcome.** We also note that such a generation technique is effective, as adding all variables requires no prior knowledge, serving as a general post-processing method for our CRRC.

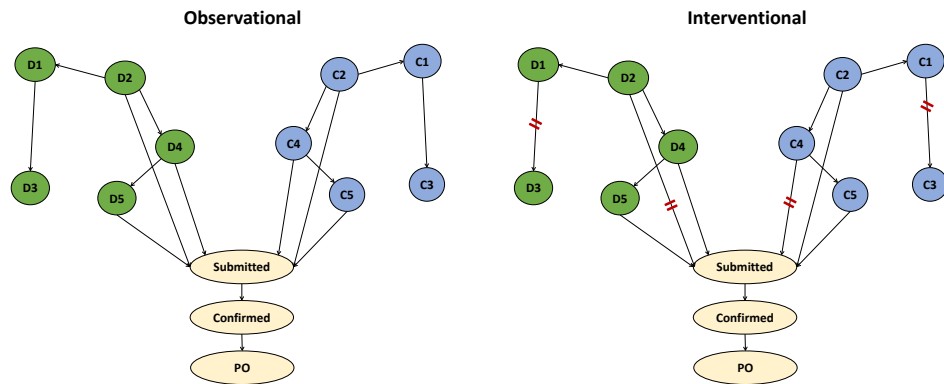

Figure 6: Illustrative Causal Graph of Semi-Supply Chain Experiment.

**Real-world Petshop Dataset.** The PetShop application (Hardt et al., 2024) is a pet adoption platform built on a microservices architecture and deployed on Amazon Web Services (AWS). Users can browse available pets and carry out adoption transactions through the system. Its functionality relies on a set of interconnected microservices—including storage components, publish–subscribe mechanisms, load balancers, and custom application logic—all containerized and orchestrated with Kubernetes. The dataset primarily aims to diagnose anomalies originating at PetSite, the front-end interface where users access the website to view pets. An anomaly is defined as a violation of the service-level objective (SLO), such as when the response time at PetSite exceeds 200 microseconds:

- **Causal Graph.** In the PetShop application, service dependencies are represented as a directed graph, where edges denote call relationships between services. By reversing the edge directions, this call graph is transformed into the causal graph used in the dataset. We use the oracle causal graph with well-specified protocols of SCM estimations in previous works when implementing the baselines (Nagalapatti et al., 2025; Okati et al., 2024) *However, our method does not acquire any prior knowledge of the causal graph during the identification process.*

- **Statistics Description.** The dataset focuses on the target node PetSite and its ancestor nodes (e.g., PetSearch ECS Fargate, petInfo DynamoDB Table, payforadoption ECS Fargate, lambdastatusupdater Lambda Function, and petlistadoptions ECS Fargate), for which KPIs such as request counts and latency metrics were collected via Amazon CloudWatch at 5-minute intervals. In total, 68 issues were injected across five nodes, each mapped to a unique ground-truth root cause, producing distinct test cases. These issues include request overloads, memory leaks, CPU hogs, misconfigurations, and artificial delays, all of which trigger SLO violations at PetSite through abnormal traffic surges. The dataset defines three categories of root cause cases—Low, High, and Temporal latency—averaging 485, 690, and 571 requests per second, respectively. While High and Temporal cases deviated strongly from training patterns and were effectively handled by baseline traversal methods, the more subtle Low latency cases posed greater challenges, where IDI achieved the best performance.

G.3   ROBUSTNESS AGAINST THE KEY ASSUMPTION 1

Our key assumption, i.e., Assumption 1 in the main paper, states that when $X_l$ is intervened, the effect from all of its ancestors $W_{jl}$ will vary. However, we perform a deeper insight against such assumption. During the proof of our identifiability analysis, the key role of Assumption 1 serves to judge whether a node $X_k$ is intervened or is not intervened but owning intervened ancestors. However, we observe that such assumption can be relaxed to a much more mild condition that:

> For any intervened $X_l$, the affect from its parental variables, i.e., $G_{ij}$ for any $X_j \in Pa(X_l)$ should vary.

To verify our insight, we further tune the ratio of varied ancestors of each intervened $X_l$ from $0.1$ to $0.9$ respectively. Results align with our relaxed version, informing that the performance of our proposed CRRC only degrades when the parental effect start to remain.

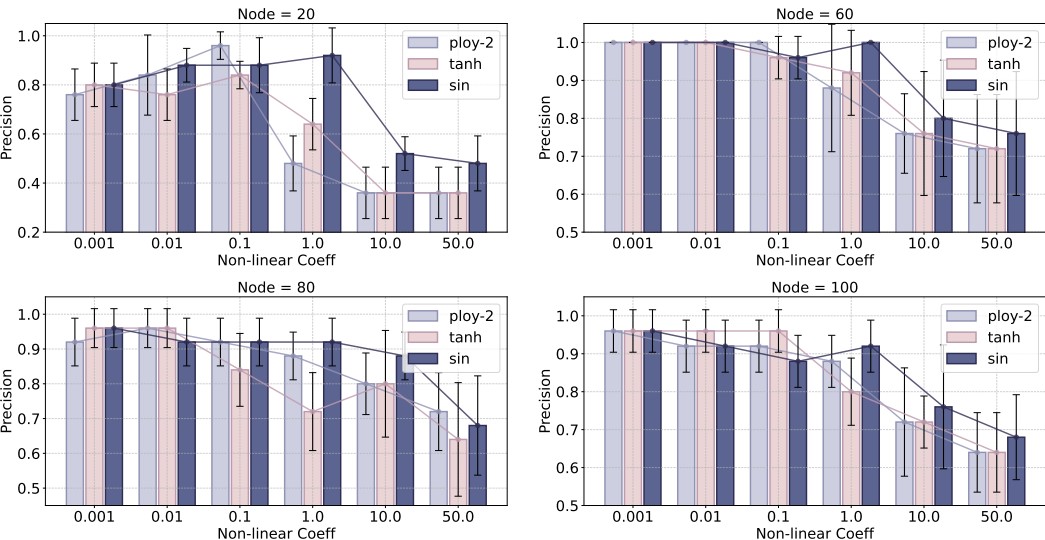

Figure 7: Robustness Against Non-linear Data Structure with different size of causal graph.

### G.4 ROBUSTNESS AGAINST THE LINEAR ASSUMPTION

One might also doubt that whether the linear SCM assumption heavily affects our method's robustness against the potential risk. Correspondingly, we introduce the non-linearity with three non-linear functions $f \in$ ["ploy-2", "tanh", "sin"]. To be specific, we add non-linear interactions and non-linear affects from the parental set of each node $X_l$ as follows:

$$X_l = \text{coffe} * (f(Pa(X_l)) + f(X_k * X_g)) + \epsilon_l, \qquad (40)$$

where $X_k, X_g$ are randomly selected from $Pa(X_l)$, and the term coffe controls the effect of the non-linearity. The larger coffe is, the stronger non-linearity is introduced. We tune the non-linear coefficient in the range of $[0.001, 0.01, 0.1, 1.0, 10.0, 50.0]$, and the full results are presented in Fig. 7.

Table 6: 6 True root causes + additional spurious ancestors (SAs)

| Ratio of additional SAs | 14.3% | 25% | 33.3% | 40% | 45.5% |
|---|---|---|---|---|---|
| Precision | $0.93 \pm 0.03$ | $0.93 \pm 0.03$ | $0.93 \pm 0.03$ | $0.93 \pm 0.03$ | $0.93 \pm 0.03$ |

Table 7: 6 True root causes - missing true ancestors (TAs)

| Ratio of missing TAs | 16.6% | 33.3% | 66.7% |
|---|---|---|---|
| Precision | $0.81 \pm 0.01$ | $0.72 \pm 0.09$ | $0.48 \pm 0.12$ |

### G.5 ROBUSTNESS AGAINST ANCESTOR DISCOVERY

Our method assumes that the set $An(Y)$ is correctly identified by Algorithm 1, while empirical estimates might exhibit some error, which can be categorized into two types. To empirically test the

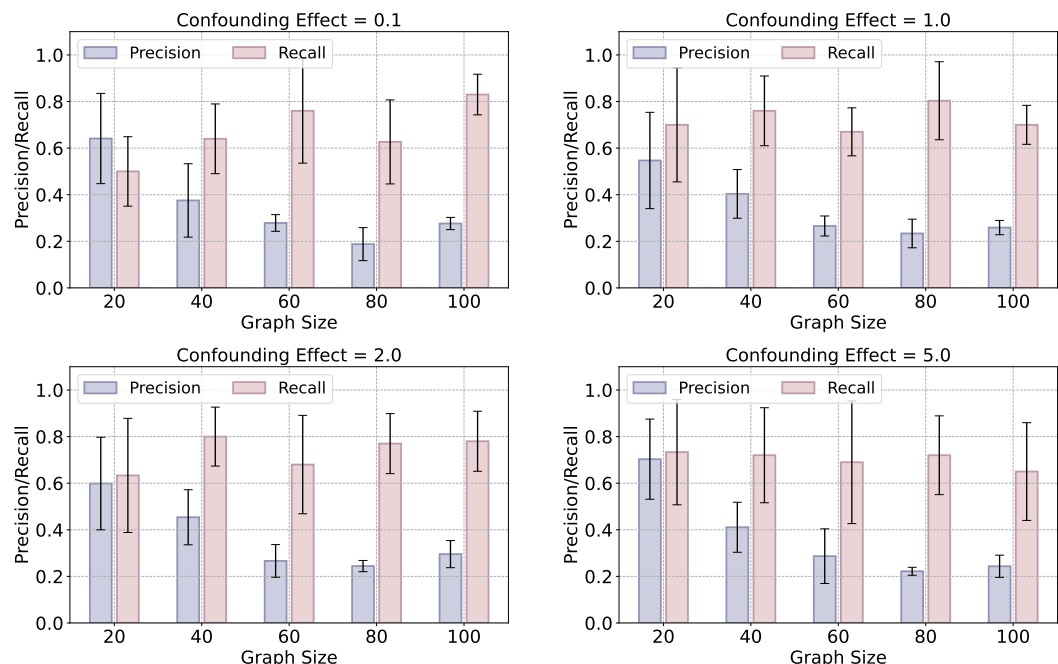

Figure 8: Robustness Against Causal Insufficienct Data Structure with different confounding effect.

robustness of our CRRC towards such error, we perform experiments by manually tuning the result of ancestor discovery, i.e., adding spurious ancestor or deleting true ancestors, in our simulation studies with $p = 80$:

- **Miss some true ones:** When true root cause is not included into the discovered ancestors, obviously CRRC would never test it and thereby miss it. Our method can not solve this problem and suffer from performance degrade when the error occurs (see Table 6).

- **Spurious ancestors:** When ASD includes a non-ancestor by mistake, this can be further categorized into two finer grained scenarios: (1) If causal sufficiency assumption hold, the non-ancestor can be excluded by our CRRC method. This is because the coefficient of it and its ancestor stays around zero before and after the intervention (see Table 7); (2) If causal sufficiency assumption does not hold and a variable confounded with Y is labeled as ancestor, CRRC cannot exclude this kind of false root cause. Our results in Appendix E.5 demonstrate this; (3) Specifically, when the confounding effect becomes larger, the precision metric of CRRC is deteriorated.

## G.6 ROBUSTNESS AGAINST THE CAUSAL SUFFICIENCY ASSUMPTION

Besides, the causal sufficiency is a very common protocols serving in the area of RCA, as the system manager can control as many hidden confounder as possible. However, one cannot guarantee whether hidden confounders will not exist in every realistic example. Hence, to test the robustness of our proposed CRRC towards the violation of causal sufficiency assumption, we introduce the hidden confounding effect in our synthetic dataset. To be specific, we first select all the confounders and perform intersection operation with the set of the root variable set in the causal graph. Then we mask all of such confounder variables during the inference stage of our CRRC, by setting the confounding coefficient of these masked confounders as the term "Confounding Effect". The larger "Confounding Effect" is, the causal sufficiency assumption will be twisted more seriously. Full results are presented in Fig. 8, where we adopt two metrics, i.e., precision and recall, to measure whether: (a) our method captures non-redundant RCs; (b) our method captures all underlying RCs, respectively. Experimental results reflect that: (1) our CRRC will capture redundant RCs due to

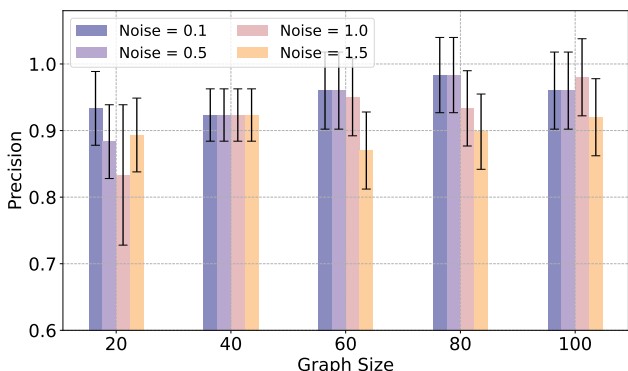

Figure 9: Robustness Against Increasing Exogenous Noise Extent.

the existence of hidden confounders, and such redundantly identified RCs increases with increasing graph size (i.e., increasing number of confounders); (2) our CRRC still captures nearly most of the underlying RCs (high recall metric across diverse cases).

### G.7    ROBUSTNESS AGAINST THE INCREASING NOISE

Moreover, we test the performance of our proposed CRRC method against increasing variance of the exogenous noise of each variables, which might migrate the local intervention of the mechanism connecting each variable with each other. To be specific, we tune the variance ranging in [0.1, 0.5, 1.0, 1.5] and record the precision of our CRRC under different graph size. Results in Fig. 9 reflect that our proposed method remains good robustness against increasing exogenous noise.

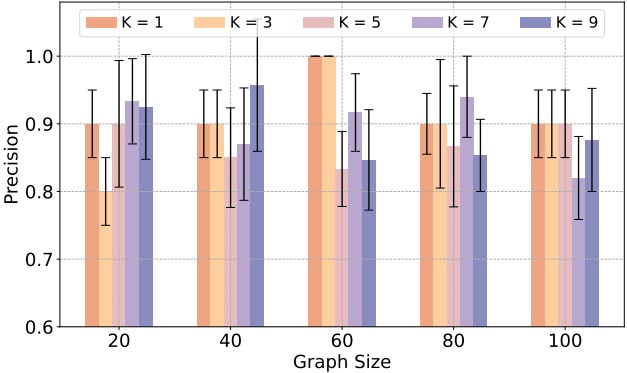

Figure 10: Robustness Against Increasing Number of the RC set.

### G.8    ROBUSTNESS AGAINST THE VARYING SIZE OF ROOT-CAUSES

Moreover, we test the performance of our proposed CRRC method against increasing variance of the exogenous noise of each variables, which might migrate the local intervention of the mechanism connecting each variable with each other. To be specific, we tune the variance ranging in [0.1, 0.5, 1.0, 1.5] and record the precision of our CRRC under different graph size. As shown in Fig. 10, our proposed CRRC achieves stable identification of the underlying RCs with varying number of the set of RCs (over $80\%$).

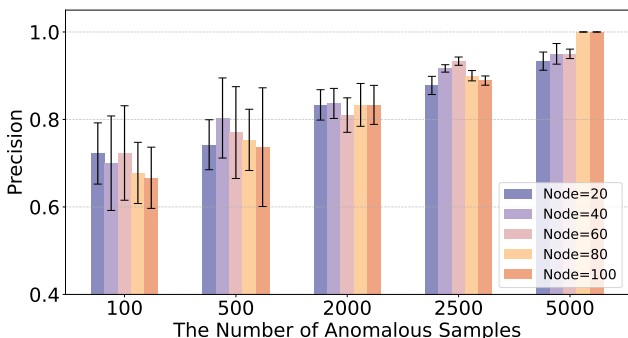

Figure 11: Robustness Against Decreasing Number of Anomalous Samples.

### G.9    ROBUSTNESS AGAINST THE VARYING SIZE OF ANOMALOUS

We then perform analysis on the sample efficiency of the proposed method. We note that one limitation of our proposed method falls in the requirement of certein number of sample when outlier happens (e.g., to perform OLS regression). Hence, to test how our proposed CRRC perform when the number of anomalous samples decreases, we tune the number of anomalous samples in the range of $[100, 500, 2000, 2500, 5000]$, and report the averaged results in Fig. 11. We observe that once the normal/abnormal samples ratio is larger than 10, the performance of our CRRC achieves ideal RC identification over $80\%$.

### G.10    ROBUSTNESS AGAINST THE VARYING CONNECTING DENSITY OF THE UNDERLYING CAUSAL GRAPH

Finally, we also test the performance of each baseline in our synthetic setting with varying connection density. To be specific, we choose the ER protocols to generate the random DAG, with the connection density denoting the total connections in the graph. Intuitively, the smaller density implies the much easier RC identification, as sparse structure of the causal graph reduces the difficulty of distinguishes RCs from non-intervened ancestors and intervened non-ancestors. As shown in Tab. **??**, our proposed CRRC achieves substantial improvement compared to each baseline across different density settings.

