# OpenReview forum: "Identifying Outcome-Oriented Root Causes via Cross Regression"
_ICLR.cc/2026/Conference — Submitted to ICLR 2026_

### Official Review · Reviewer_ryEZ · 2025-10-24

**Soundness:** 2
**Presentation:** 2
**Contribution:** 2
**Rating:** 2
**Confidence:** 4

**Summary:**

The paper presents a root cause analysis algorithm that identifies all ancestral nodes of an outcome variable. It is done by cross-regressing the observational and interventional data on the outcome variable. The paper also provides an identifiability result of the root causes for the proposed algorithm.

**Strengths:**

- Under linear SCM, the paper uses an existing result to develop an efficient RCA method given a target variable.
- It presents the main algorithm and the core idea clearly.
- The experiment involves a wide range of algorithms. It also analyzes different aspects of the RCA methods.
- The proposed method does not assume any causal graph or SCM given as a priori.

**Weaknesses:**

- The paper cites a paper that was under review on openreview from ICLR 2025 but does not cite a relevant paper that has already been published by Ikram et.al 2025.
- The algorithm is not applicable in cases where an outcome variable is not given as a prior when a failure happens.
- The motivation of the paper is weak and the paper also show signs of unfamiliarity of the existing work. The paper claims that existing RCA methods aim to identify the set of all intervened nodes instead of only the intervened nodes that affect a particular outcome. Besides Budhathoki et al., 2022; Okati et al., 2024; anonymous, 2025, some methods that do not assume a causal DAG up front, but learning a causal graph as a part of the RCA method via causal discovery. One example is CIRCA by Li at al. 2022, as cited by the paper. One can then check whether the intervened nodes are ancestors of the desired outcome variable.
- There are some very glaring grammatical issues presented in the paper. For example, line 51, which says ‘which models the happens of anomaly as interventions…’.
- The index notation used for representing random variables are confusing in section 2.
- The proposed model assumes the data generative mechanism to be a linear SCM, which allows the paper to use the result from the paper about ancestor regression in the linear SEM for discovering the ancestral relations.
- The paper does not provide code for the experiment. It becomes impossible to validate the experimental results.
- The theorem numbers in the appendix do not match the main paper. For example, Theorem 2 in the main paper is actually Theorem 3 in the appendix. Proposition 1 in the main paper is actually Theorem 1 in the appendix. Theorem 1 in the main paper is Theorem 2 in the appendix. Also, line 782 says to ‘prove a simplified version of Theorem 1 as below.’ But the theorem below is called Theorem 2 (Simplified).
-	The proofs for Theorem 3 and Theorem 4 in this paper are very similar to those of Theorem 2 in the paper by Schultheiss et al. 2023, limiting the original contributions of the paper. The difference is that Schultheiss et al. 2023 focus on testing whether $\beta$ is zero, but this paper focuses on testing the difference between two beta terms. One can also see that Proposition 1 (Theorem 1 in the appendix) presented in this paper is similar to Theorem 1 in the paper by Schultheiss et al. 2023. It’s just that this paper now has an additional beta term coming from the interventional data.
-	Line 1491: the Table reference number is missing.
-	Line 466: double quotation marks.

Reference :

-	Ikram, Azam, et al. "Root Cause Analysis of Failures from Partial Causal Structures." The 41st Conference on Uncertainty in Artificial Intelligence.

-	Schultheiss, Christoph, and Peter Bühlmann. "Ancestor regression in linear structural equation models." Biometrika 110.4 (2023): 1117-1124.

**Questions:**

- Why is it fair to compare precision with recall in the experiment?
- Why are some of the baselines, such as RCD, iSCAN, and IDI, not compared in Figure 4?
- Why does Figure 5 (right) not include the baseline of CRRC without the discovery method?  Is it CRRC (+AREG)?
- Can CRRC handle the case when there are multiple outcome variables of interest?
- About Table 3:
    - How come the performance of IDI and Traversal reported are different than what has been reported in Table 1 by Nagalapatti 2025 when the code and dataset are from their paper?
   - How can RCD get 0 recall for k=3 but 0.07 recall for k=1 under ‘high’ category?
- About experimental details in Appendix G
     - At line 1197, what does ‘sample the adjacency matrix entries from a uniform distribution on $[-2,0) \cup (0, 2]$’ mean? Also, how does it reconcile with the expected outgoing edge size as 100 in SF graphs or the expected neighborhood size as 2 in ER graphs, since they are all about adjacency in the graph?
     -  At line 1203,  what does ‘from $[-2,0) \cup (0, 2]$ to $U([-1,1) \cup (0, 2])$ mean? It seems to be related to what ‘sample the adjacency matrix entries from a uniform distribution on $[-2,0) \cup (0, 2]$’ means from my previous question. Also, isn’t $ U([-1,1) \cup (0, 2]) = U([-1, 2])$?
     - At line 1206, will the last int(K/4) ancestors be descendants of top int(K/4)? If so, how come the nodes between them are not considered as ground truth root causes? Are there any real-world practical scenarios that support this setup?
    - Are the top int(K/4) nodes source nodes (nodes that have no incoming edges) or just ancestors of the outcome? If they are only ancestors, can the authors justify why it is necessary to know the descendants of the source node as the root causes?
     - At line 1211, it says the overall sample size is 2000. How many samples are interventional samples?

---

### Official Review · Reviewer_EASa · 2025-10-27

**Soundness:** 3
**Presentation:** 1
**Contribution:** 2
**Rating:** 4
**Confidence:** 3

**Summary:**

The paper introduces that new root cause analysis method called CCRC that instead of finding all intervened variables, it targets intervened ancestors of the outcome. The proposed method first identifies ancestors of the target via "ancestor regression", then performs a cross-regression test by the replacing the k-th feature column with its interventional counterpart and comparing OLS coefficients between observational and "cross-replaced data". The authors prove identifiability under a "regularity on ancestor weight matrix" assumptions drive asymptotic normality for coefficient differences to justify the hypothesis tests, and report strong empirical results.

**Strengths:**

* Clear problem framing: focusing on intervened ancestors of a target Y instead of finding all intervened variables

* Simple scalable procedure

* Theoretical guarantees

* Robustness checks

**Weaknesses:**

* Several claims in the paper are presented without sufficient intuition or explanation of why they should hold.

* Assumption 1 appears strong and it is not thoroughly  discussed.

* Replacing a column with interventional values implicitly constructs a hybrid sample mixing observational X-k and interventional data Xk. It seems to be that this ignores induces dependencies between  Xk and X-k under intervention. The theorticical section should more explicitly justify why the OLS coefficient mapping works under such hybrid data.

* Although the problem formulation is clearly described, its practical motivation is not well established. It is unclear in which concrete applications root-cause analysis would be carried out exclusively with respect to a single target variable

* The experimental section lacks clarity, particularly regarding the setup and interpretation of the real-data experiments.

* Several state-of-the-art baselines are missing, and those included are not discussed in sufficient detail. In general, I think related  works should be discussed more thoroughly in order to understand the contribution of this paper regarding root cause analysis.

* The experimental results are surprisingly strong; however, some established methods that typically perform well show unexpectedly poor results (like RCD) under the paper’s setup. It would have been helpful to have access to the code or supplementary material to better understand these discrepancies.

**Questions:**

*  It is not clear to me why the authors assume a linear SCM and then use a nonlinear function f in lemma 1. Although the lemma was established in prior work, providing some intuition for readers would be very helpful. In particular, could the authors explain why this regression identifies ancestors specifically, rather than the Markov boundary of Y?

* If multiple ancestors are intervened simultaneously, can coefficient cancellations occur (masking detection)? Any diagnostic for such cases?

* Is it possible to explain why hybrid data preserves the coefficient relation? Would a two-sample regression or causal score test avoid hybridization yet support similar guarantees?

* In what common intervention scenarios does Assumption 1 hold? Is it possible to provide counterexamples where it fails?

* Traditionally, root-cause analysis seeks to identify all causes of anomalies in a system. Could the authors give a concrete real-world example where it is meaningful to focus on anomalies affecting only a specific target variable, rather than all anomalies in the system? In words, can the authors think of an application where only the restoration of a single outcome is of practical interest, and other anomalies are acceptable as they are and do not need to be eliminated.

 * Can the authors explain why their methods perform well in situations violating causal insufficiency. Is that that a weaker version of  causal sufficiency is sufficient for the algorithm. Or is it that the experimental design focused on situations where violating causal sufficiency is not a real issue?


* If I understood correctly, i-SCAN [2] was not introduced for root cause analysis. Can the authors explain how they adapted it for RCA?

* As far as understood, CIRCA needs a graph to be specified. If that is the case, this should be clarified and I think the auhtos should explain how they specified the graph? Other methods that do not require a graph should be included. Like for example the method proposed by [3].

* As far as I know, the dataset provided by [4] is not a dataset containing anomalies and root causes. Even after looking at the appendix I do not understand how anomalies are detected or even if there are anomalies. I recommend using datasets introduced for anomalies and root cause analysis. Like for example the one introduced in [5].

* Several of the methods that authors compared to, are not suited for the experimental design. For example, RCD, as far as I understood, does not assume a change in the causal coefficient in the SCM, instead it assumes a change in the noise. I wonder how good will their method perform against closer methods that also compare different in coefficients in normal regime and in the anomalous regime? For example, as it as done in [5]?



Minor:

* Maybe I'm missing something, but isn't ANS(Y) = An(Y)?

* Provide a small worked example showing the coefficient-comparison logic step-by-step.

* The statement "We allow for X in Pa(X) and X in An(X)" might confuse some readers, as it could be interpreted as allowing cycles in the graph. I would suggest rephrasing this to make it explicit that the authors are simply following the convention that a variable is considered a parent and ancestor of itself. In the current formulation, it appears that a vertex can be its own parent in some cases but not necessarily always, which may lead readers to think that this property depends on the specific graph rather than being a fixed definitional convention.


References:

[1] Schulthesiss and Buhlmann. Ancestor regression in linear structural equation models. Biometrika. 2023

[2] Chen, Bello, Aragam, Ravikumar. iscan: identifying causal mechanism shifts among nonlinear additive noise models. Neurips. 2024

[3] Meng, Zhang, Sun, Zhang, Hu, Zhang. Localizing Failure Root Causes in a Microservice through Causality Inference. IEEE ACM IWQoS. 2020

[4] Sachs, Perez, Peer, Lauffenburger, Nolan.  Causal proteinsignaling networks derived from multiparameter single-cell data. Science. 2005

[5] Assaad, Ez-Zejjari, Zan. Root Cause Identification for Collective Anomalies in Time Series given an Acyclic Summary Causal Graph with Loops. AISTATS. 2023

---

### Official Review · Reviewer_VgX3 · 2025-10-30

**Soundness:** 3
**Presentation:** 2
**Contribution:** 2
**Rating:** 4
**Confidence:** 4

**Summary:**

This paper attempts to identify the root causes of anomalies. In an interconnected system, the authors model the root cause as nodes that have undergone a (soft) intervention and have a causal effect on the outcome variable \( Y \), thereby causing an anomaly at \( Y \). The main strength of this paper is that it addresses this problem without requiring a causal graph. The authors build upon prior methods that identify ancestors of a node via hypothesis testing, and then search for root causes among the ancestors of \( Y \).  To find the root causes, they assume a linear Structural Causal Model (SCM) and compare the regression coefficients obtained before and after intervention. They then use hypothesis testing to determine whether the intervention indeed caused the anomaly at \( Y \).

**Strengths:**

1. The problem formulation that avoids requiring a causal graph is a notable strength.
2. The experiments are extensive, and the coverage of baseline methods is adequate.
3. Although a linear DGP is assumed, the results appear strong.

**Weaknesses:**

1. Code is missing. I request the authors to release it along with clean installation and running instructions. I assure the authors that the reviewer will attempt to reproduce some results before arriving at a final decision.
2. The paper needs more polishing. It appears that the paper was written in haste, leading to a suboptimal presentation of ideas.
3.  The paper makes strong assumptions that restrict the scope in which their method is useful (elaborated below)

**Questions:**

1. Lines 13–17 and 54–58 are misleading and need revision. Several causal RCA approaches including TOCA, CF-Attn, and IDI have already emphasized that the root cause should be nodes that are not only intervened but also cause the anomaly at \( Y \) as a consequence of the intervention. The narrative should be corrected accordingly.
2. Figure 1(c) needs revision in light of comment (1).
3. At line 105 and throughout the paper, please use bold-faced capital \( \mathbf{X} \) for matrix notation.
4. A major weakness is the assumption of a linear DGP (lines 111–114). Having this restriction at the problem formulation stage seems overly limiting.
5. The authors should justify their choice of using soft interventions beyond citing prior work. Specifically, please discuss why a causal mechanism (structural equation) would change as a result of an intervention. A real-world example illustrating such a structural change would strengthen the motivation.
6. Another significant weakness is the assumption of two sources of paired data — \( D \) (normal observational data) and \( \tilde{D} \) (post-intervention data). Although the authors refer to \( \tilde{D} \) as interventions, I believe that this paired-data assumption effectively requires counterfactual instances, which is a very strong assumption in causal inference and further restricts the problem scope.
7. In view of (6), if the authors still believe counterfactual data is not necessary, please justify the cross-regression step where the \( k \)-th attribute in \( D \) is replaced with the \( k \)-th attribute from \( \tilde{D} \).
8. Can the authors elaborate on how Lemma 1 helps distinguish correlation from causation, which is crucial for identifying ancestors? I know that some approaches like LinGAM assume non-Gaussian exogenous noise and exploit asymmetry of error residuals to identify ancestors. However, the authors do not make such assumptions yet claim to identify ancestors—please clarify.
9. For each dataset in the results section, please also report the accuracy of ancestor identification for \( Y \) to provide better context for the reported results.
10. At line 173, why was \( f = Y^3 \) chosen? I understand that prior work [Christoph Schultheiss and Peter Buhlmann] used this function, but please provide motivation for why it is appropriate for your setting.

---

### Official Review · Reviewer_9ih8 · 2025-11-01

**Soundness:** 3
**Presentation:** 3
**Contribution:** 3
**Rating:** 6
**Confidence:** 3

**Summary:**

The paper considers a problem that is related to root-cause analysis: identify the intervened _ancestor_ variables of the outcome. They propose a method and evaluate it against a wide set of prior methods. The method obtains accuracy gains on synthetic datasets.

**Strengths:**

- The proposed method does not require access to a causal graph, unlike prior approaches using counterfactuals.
- Significant improvement in accuracy compared to prior work.
- An extensive set of ablations that study the robustness of the method, e.g., to causal sufficiency assumption.

**Weaknesses:**

- Significant gains are observed only on synthetic datasets. On the (semi)real-world PetShop dataset, the method is only slightly better than existing methods such as IDI.
- writing clarity can be improved (e.g., Table 1 caption mentions, "Precision or recall@k"--which one is being reported?)
- While the novelty of restricting to ancestors is motivated as a key contribution, the solution simply uses a prior method for identifying ancestors.

**Questions:**

- How important is the choice of non-linear f? Can you provide some ablations on its choice.
- Why is the improvement of recall marginal for PetShop?
- Is there a way to include a ranking over the identified ancestors? Identifying all may not be practically useful.

---

### Meta-Review · Area_Chair_7yYs · 2025-12-30

**Summary:**

This paper proposes a method for root cause analysis.
The reviewers raised concerns about the clarity of the paper and the lack of relevant lieratures.

**Reviewer Concerns:**

There were no rebuttals submitted for this paper.

**Reviewer Scores:**

Because there were no rebuttals, I believe the reviewers would have kept their original scores.

---

### Decision · Program_Chairs · 2026-01-26

Reject